# Evaluation of soil moisture from CCAM-CABLE simulation, satellite based models estimates and satellite observations: Skukuza and Malopeni flux towers region case study

Floyd V. Khosa[1,2], Mohau J. Mateyisi[1], Martina R. van der Merwe[1], Gregor T. Feig[1,3,4], Francois A. Engelbrecht[5,6], Michael J. Savage[2]

*Correspondence to*: Floyd Khosa (vukosikhosa@yahoo.com)

[1] CSIR, Natural Resources and the Environment – Global Change and Ecosystem dynamics, P.O. Box 395, Pretoria 0001, South Africa

[2] Agrometeorology Discipline, School of Agricultural, Earth and Environmental Sciences, University of KwaZulu-Natal, Pietermaritzburg, South Africa

[3] Department of Geography, Geoinformatics and Meteorology, University of Pretoria, South Africa

[4] South African Environmental Observation Network (SAEON), P.O. Box 2600, Pretoria 0001, South Africa

[5] CSIR, Natural Resources and the Environment – Climate Studies, Modelling and Environmental Health, P.O. Box 395, Pretoria, 0001, South Africa

[6] Global Change Institute (GCI), University of the Witwatersrand, Johannesburg, 2050, South Africa

## Abstract

Reliable estimates of daily, monthly and seasonal soil moisture are useful in a variety of disciplines. However, the availability of continuous in situ soil moisture observations in southern Africa barely exists. Therefore, process based simulation model outputs are a valuable source of climate information, needed for guiding farming practises and policy interventions at various spatio-temporal scales. Despite their ability to yield historic and future projections of climatic conditions, simulation model outputs often reflect a degree of systematic uncertainty hence it is very important to evaluate their representativeness of spatial and temporal patterns against observations. To this effect, this study presents an evaluation of soil moisture outputs from a simulation and satellite data based soil moisture products. The simulation model consists of a global circulation model known as the conformal-cubic atmospheric model (CCAM), coupled to the CSIRO Atmosphere Biosphere Land Exchange model (CABLE). The satellite based soil moisture data products include satellite observations from the European Space Agency (ESA), and satellite observation based model estimates from the Global Land Evaporation Amsterdam model (GLEAM). The evaluation is done for both the surface (0-10 cm) and root zone (10-100 cm) using in situ soil moisture measurements collected from two savanna sites, located in the Kruger National Park, South Africa. For the two chosen sites, with different soil types and vegetation cover, the evaluation considers soil moisture time series aggregated to a monthly time scale from all the data sources and a standardised soil moisture index (SSI). In order to reflect the inter-comparability of CCAM-CABLE simulation output and GLEAM model estimates, a qualitative analysis of phase agreement using wavelet analysis is presented. The onset and offset of the wet period, for the two specific sites, is calculated for each of the models and the soil moisture time series

mutual information (MI) between CCAM-CABLE and the GLEAM models is discussed. The results
indicate that both the simulation and satellite observation based model outputs are generally consistent
with the in situ soil moisture observations at the two study sites, especially at the surface. CCAM-
CABLE and GLEAM inter-comparison also shows that the models are generally in phase. However,
there is a time lag of about 12 and 20 days on average, for the surface and root zone respectively. In
general, the simulation compares well with the GLEAM model estimates across different landscapes,
indicating that the key physical processes that drive soil moisture in CCAM-CABLE and GLEAM, at
the surface and root zone, lead to an appreciable degree of MI. This is reinforced by a predominantly
positive measure of MI between the respective two soil moisture outputs.

**Keywords**: atmospheric model, cross wavelet, flux tower, land surface model, soil moisture

## 1    Introduction

Accurate estimates[1] of daily, monthly and seasonal soil moisture are important in a number of fields
including agriculture (McNally et al., 2016), water resources planning (Decker, 2015), weather
forecasting (van den Hurk et al., 2012) and the quantification of the impacts of extreme weather
events such as droughts (Sheffield and Wood, 2008), heat waves (Fischer et al., 2007; Lorenz et al.,
2010) and floods (Brocca et al., 2011). Soil moisture has been identified as one of the 50 essential
climate variables (ECVs) by the Global Climate Observing System (GCOS) and the European Space
Agency climate change initiative (ESA-CCI) (McNally et al., 2016). Available soil moisture  affects
the fluxes of heat and water at the surface and directly impacts local and regional weather patterns
(Dorigo et al., 2015; Raoult et al., 2018; Yuan and Quiring, 2017). It is a key parameter to consider in
the partitioning of precipitation and net radiation. Precipitation is partitioned into evapotranspiration
(ET), infiltration and runoff.  Latent and sensible heat fluxes are components of the net terrestrial
radiation (Xia et al., 2015; Yuan and Quiring, 2017) at the surface. Root zone soil moisture plays a
vital role in the transpiration process of evapotranspiration (ET) especially in arid and semi-arid
regions, where most of the water loss is accounted for by transpiration during the dry period
(Jovanovic et al., 2015; Palmer et al., 2015). The dry period in this study refers to months when the
sites experiences minimum rainfall, which is during the austral winter period (May to October). The
temporal and spatial variation in soil moisture is controlled by vegetation, topography, soil properties
and climate variability (Xia et al., 2015).  Regions where soil moisture strongly influences the
atmosphere are at the transition between wet and dry climates. This is associated with the strong
coupling between ET and soil moisture which is a characteristic of these regions (van den Hurk et al.,
2012; Lorenz et al., 2010).

In situ data that are used as a reference in this study consist of surface and root zone soil moisture
observations. The in situ data are mostly point based, which poses significant challenges  in
understanding the spatial patterns in soil moisture (Yuan and Quiring, 2017). Direct satellite
observations, on the other hand, are presently only available for the surface. To obtain root zone
estimates of soil moisture satellite based surface soil moisture data are used in conjunction with
ground-based observations and model estimates. The modelled soil moisture data are largely
dependent on accurate surface forcing data (e.g. air temperature, precipitation and radiation) and the
parameterisation of the land surface schemes (Xia et al., 2015). This is done in the frame work of
physically based models whose accuracy may vary depending on the response of the models to the
forcing data. Due to lack of publicly available long term and complete in situ soil moisture

---

[1] *Estimate* here refers to both process based model simulation and satellite derived data products thereafter, the term
*simulation* will be used for process based model outputs while, *estimates* will be reserved for satellite derived data.

measurements in South Africa, and the world in general, global climate models (GCMs) are relied on to estimate the land surface states (Dirmeyer et al., 2013). The data produced by land surface models, hydrological models and GCMs have been widely evaluated for many continents and regions (Albergel et al., 2012; An et al., 2016; Dorigo et al., 2015; McNally et al., 2016; Yuan and Quiring, 2017). The evaluations of these soil moisture data products in Africa are sparse, mainly due to the lack of publicly available in situ observations (Sinclair and Pegram, 2010). The available studies include those conducted by McNally et al. (2016) and Dorigo et al. (2015a) evaluating ESA-CCI satellite soil moisture products over east and west Africa, respectively. This study is inspired by the notion that the knowledge about soil moisture characteristic patterns, for the study region, can reliably be obtained by making a connection between data from simulation experiments, theoretical or analytical models and in situ observations. Satellite and model based soil moisture products are capable of providing continuous observations at different temporal and spatial resolutions (Fang et al., 2016). Despite the in situ data being limited in coverage, they are very useful for the calibration and validation of modelled and satellite derived soil moisture estimates (Xia et al., 2015).

The aims of this study are twofold. Firstly, it is to evaluate the ability of the model simulated and satellite derived soil moisture products to capture the observed variability in soil moisture at specific locations. The evaluation is undertaken at two soil depths namely; surface (SSM, i.e., 0-10 cm) and root zone (RZSM, i.e. 10-100 cm), using long term in situ measurements. The evaluation is limited to two study sites that are located in the Kruger National Park in South Africa. This is due to limited publicly available in situ measurements within the study area of north-eastern South Africa. For example, the global flux data network (FLUXNET) has only one a single data site listed in this region (which is the flux tower at Skukuza) and the International Soil Moisture Network (ISMN) with no data sites in the study region. Secondly, the goal is to inter-compare climate model simulated results of soil moisture against satellite-based estimates. This is done primarily at a regional level, where the absence of sufficient in situ observations over space and time presents a major challenge for climate model verification. In particular, we look at spatio-temporal variations in simulated soil moisture data from a coupled land-atmosphere model i.e., the conformal cubic atmospheric model (CCAM) of the Commonwealth Scientific and Industrial Research Organisation (CSIRO) coupled to the CSIRO Atmosphere Biosphere Land Exchange (CABLE) model against the three versions of the European Space Agency (ESA) satellite observations (i.e., active, passive and combined), and estimates from three versions of the global land evaporation Amsterdam model (GLEAM). The goal is to investigate how the process based CCAM-CABLE simulations, satellite derived soil moisture observations and GLEAM model estimates compare with the in situ observations, in capturing the seasonal cycles of soil moisture at a point. In addition, the extent to which the climate model simulations and GLEAM model estimates have MI at the regional level within inter-annual time scales is investigated. In particular, the aim is to investigate phase agreement between the respective soil moisture data products and establish if they are representative of local conditions. The study seeks to uncover interesting patterns in the observed data, for the study region, and highlight the strengths as well as aspects of the climate model simulation and GLEAM model estimates which may benefit from continuous testing and improvement.

Clearly the ability of models to capture seasonal cycles of terrestrial processes such as soil moisture is indicative of how well the physical processes that underlie the variability of soil moisture over space and time are represented. A comparison of satellite-derived products with in situ observations may also yield useful insight on the strengths and weaknesses of various remote sensing techniques that are used. A climate models' ability to represent and capture the seasonality of a system under inter- and intra-annual climate variability could be considered more important than its agreement with

observations in absolute values (Fang et al., 2016). The remainder of study is structured as follows: Sect. 2 describes the datasets used, the study design and analysis of the datasets. Section 3 presents the results and the discussion, followed by the conclusions in Sect. 4.

## 2    Materials and methods

### 2.1    Study sites and in situ observations

In situ soil moisture measurements from the Council for Scientific and Industrial Research (CSIR) operated network of eddy covariance flux towers, in the Lowveld region of the Mpumalanga (Skukuza) and Limpopo (Malopeni) provinces are used. A number of other soil moisture in situ measurements sites exist in the country, however, their data are not publicly available.

### 2.1.1    Skukuza

The Skukuza flux tower site is a long-term measurement site, located within the Kruger National Park conservation area in South Africa (25.0197° S, 31.4969° E; Fig. 1). The Skukuza flux tower has been operational from 2000 to present. The site falls within a semi-arid savanna biome at an altitude of 370 m above sea level, with a mean rainfall of 547 mm year$^{-1}$, and average annual minimum (during the dry season) and maximum (during the wet season) temperatures of 14.5 and 29.5˚C, respectively for the averaging period from 2001 to 2014. The vegetation is dominated by an overstory of *Combretum apiculatum* (Sond.), and *Sclerocarya birrea* (Hochst.) with a height of approximately 8-10 m, and a tree cover of approximately 30% (Archibald et al., 2009). The understory is a grass layer dominated by *Panicum maximum* (Jacq.), *Digitaria eriantha* (Steud.), *Eragrostis rigidor* (Pilg.) and *Pogonarthria squarrosa* (Roem. and Schult.). The soil is of the Clovelly form with a sandy loam texture (Feig et al., 2008), and the dominant soil type for the 25 km resolution grid cell where the flux tower is located is silty loam (Fig. 1). The Skukuza flux tower site is extensively described in previous studies including those by Archibald et al. (2009) and Scholes et al. (2001). In situ soil moisture data are collected 90 m north of the tower, and the measurements are taken at two profiles which are 8 m apart. The sensors are located at four different depths for both profiles i.e., 5, 15, 30 and 40 cm (Pinheiro and Tucker, 2001). Time domain reflectometry (TDR) probes (Campbell Scientific CS615L) are used to measure soil moisture at a 30-minute temporal resolution. These measurements were averaged to a daily time period (using an 80% data threshold) to match the resolution of the other soil moisture products. In this study the in situ data from the year 2001 to 2014 are used.

### 2.1.2    Malopeni

The Malopeni flux tower is located 130 km north west of the Skukuza flux tower (23.8325° S, 31.2145° E; Fig. 1), at an elevation of 384 m above sea level. The tower has been collecting data since 2008 to present, however data was not collected between January of 2010 and January of 2012 due to equipment failure. The site has a mean rainfall of 472 mm year$^{-1}$, and annual average minimum and maximum air temperatures of 12.4 and 30.5˚C respectively, for the averaging period from 2008 to 2014. The site is dominated by broad leaf *Colophospermum mopane*, which characterise a hot and dry savanna (Ramoelo et al., 2014), *Combretum apiculatum* and *Acacia nigrescens* are also abundant at the site. The grass layer is dominated by  *Schmidtia pappophoroides* and *Panicum maximum*. The soil at the site is predominantly of the shallow sandy loam texture, and the dominant soil type for the 25 km resolution grid cell where the flux tower is located is silty loam (Fig. 1). The soil moisture probes are located at four different profiles and depths. The sensor types and depths positioning are the same for the Malopeni and Skukuza flux tower sites.  Soil moisture is collected at four different profiles

(i.e., 16 sensors at four depths) and averaged to represent surface and root zone soil moisture at this
site.

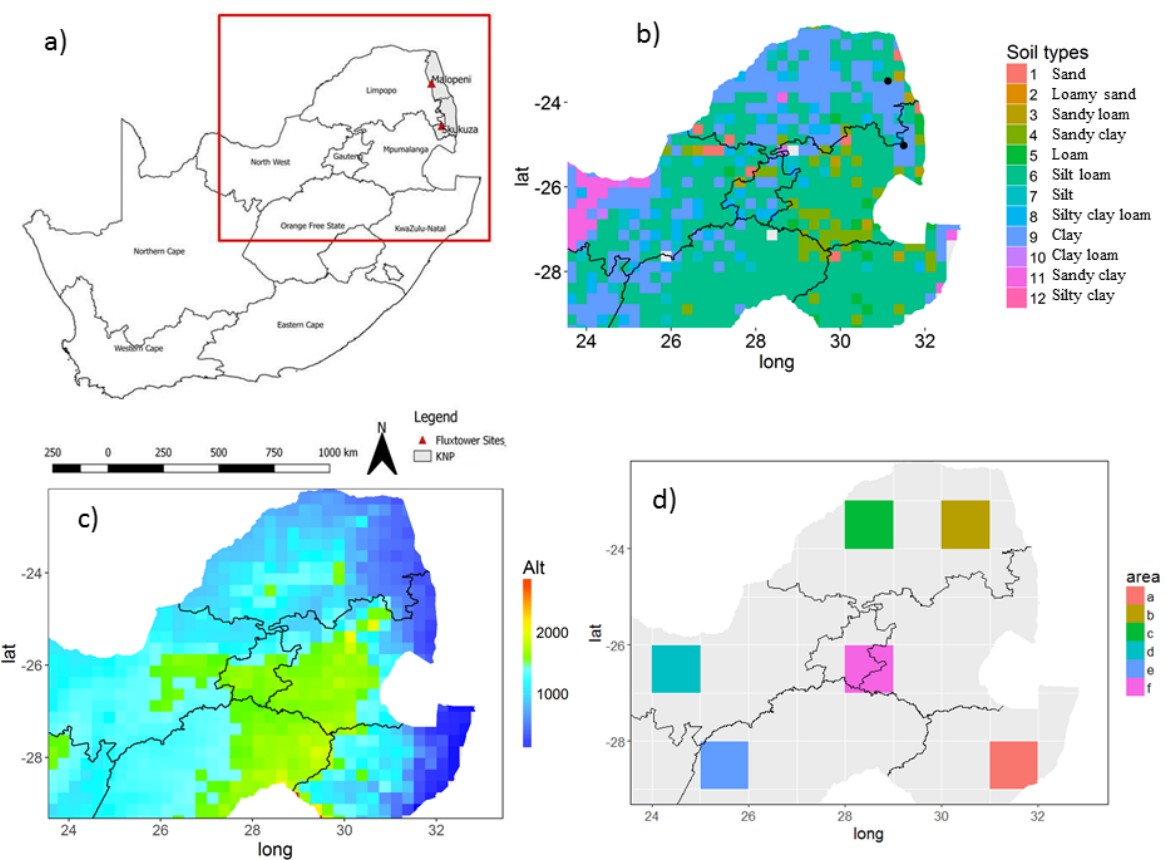

**Figure 1.** Maps indicating (a) South Africa, Kruger national park (KNP), flux tower sites (Skukuza and Malopeni) and the area considered for grid inter-comparison (red box), (b) dominant soil types per grid cell, at a resolution of 25 km, (c) the altitude (Alt, m) at the study region and (d) regions selected by increasing altitude anti-clock wise (i.e., a-f) and hydrological zones.

## 2.2   Satellite observations

The European Space Agency climate change initiative (ESA-CCI) satellite-derived soil moisture datasets are used in this study (Liu et al., 2012; Yuan and Quiring, 2017). These global datasets are based on passive and active satellite microwave sensors, and provide surface soil moisture estimates at a resolution of ~25 km (i.e., 0.25˚) (Fang et al., 2016; Yuan and Quiring, 2017). The ESA-CCI merges soil moisture estimates from the active and passive satellite microwave sensors into one dataset (http://www.esa-soilmoisture-cci.org/), using the backward propagating cumulative distribution function method (Dorigo et al., 2015; Fang et al., 2016). A detailed description of the merged active and passive sensors and their individual functioning is provided by Fang et al., (2016), Dorigo et al., (2015) and Liu et al., (2012). The merging of active and passive sensors is based on their sensitivity to vegetation density, as the accuracy of these products varies as a function of vegetation cover (Liu et al., 2012). In this study, version 3.2 (v3.2) of the ESA-CCI soil moisture data is used.

A number of studies evaluated these products at a regional and global scale using in situ data and concluded that passive sensors displayed improved performance over bare to sparsely vegetated

regions, whereas the active sensors perform better in moderately vegetated regions (Al-Yaari et al., 2014; Dorigo et al., 2015; Liu et al., 2012; McNally et al., 2016). Over densely vegetated areas such as tropical forests, neither product produces reasonable estimates. The dense canopy hinders signals reflected from the soil surface (i.e. for passive sensors) or back scattering of active radiation before it reaches the soil surface for active sensors (Liu et al., 2012). The merged data product is used in this study as it has better data coverage compared to the individual products. Missing data in satellite products are not unusual since retrievals are normally at an interval of 2-3 days (Albergel et al., 2012). However, data from each of the different sensor types are also individually considered for the evaluation of long-term seasonal cycles.

## 2.3 Models for simulating soil moisture

### 2.3.1 CCAM-CABLE

The variable-resolution atmospheric model CCAM developed by the CSIRO in Australia (McGregor, 2005; McGregor and Dix, 2001, 2008) was used to dynamically downscale ERA reanalysis data to 8 km resolution over north-eastern South Africa for the period 1979-2014. The domain was centred over the Waterberg in South Africa and over a region of about 1500 x 1500 km$^2$. Spectral nudging of the CCAM atmospheric simulations in the ERA reanalysis data took place through the application of a digital filter using a 600 km length scale. The filter was applied at six-hourly intervals and from 900 hPa upwards. Similar downscalings of reanalysis data obtained over southern Africa using CCAM are described by Engelbrecht et al., (2011), Dedekind et al., (2016) and Horowitz et al., (2017).

CCAM was integrated coupled to the dynamic land-surface model CABLE (Kowalczyk et al., 2006a) in order to perform the simulations. The CCAM-CABLE model outputs were stored at six-hourly time-resolution with daily maxima of a number of variables also being stored. The radiative forcing of the simulations, including $CO_2$ and ozone concentrations were obtained from the Coupled Model Inter-comparison Project Phase Five (CMIP5). The simulations were performed on the supercomputers of the Centre for High Performance Computing (CHPC) of the Meraka Institute of the CSIR in South Africa. The ability of the CCAM model to realistically simulate present-day southern African climate has been extensively demonstrated (e.g. Engelbrecht et al., 2015, 2011, 2009; Malherbe et al., 2013; Winsemius et al., 2014).

The CABLE soil submodel expresses soil as a heterogeneous system consisting of three constituent phases namely water, air and solid (Kowalczyk et al., 2006b; Wang et al., 2011). Air and water compete for the same pore space, and the change in their volume fractions is due to drainage, precipitation, ET and snow melt. In this model there is no heat exchange between the moisture and the soil due to the vertical movement of water, as soil moisture is assumed to be at ground temperature. The soil is partitioned into six layers, with the layer thickness of 0.022 m, 0.058 m, 0.154 m, 1.085 m and 2.875 m from the top layer. Only the top layer contributes to evaporation while plant roots extract water from all layers depending on the soil water availability and the fraction of plant roots in each layer (Wang et al., 2011). Soil moisture is solved numerically using the Richard's equation (Kowalczyk et al., 2006b).

### 2.3.2 GLEAM

The Global Land Evaporation Amsterdam Model (GLEAM) version 3.1 is a set of algorithms used to estimate surface, root-zone soil moisture and terrestrial evaporation using satellite forcing data (Martens et al., 2017). The method is based on the use of the Priestley and Taylor (1972) evaporation

model, stress module, and the rainfall interception model (Miralles et al., 2011). Three data sets from the GLEAM namely v3a, v3b and v3c were used in this study. The data are freely available at www.gleam.eu. Version 3a is based on satellite observed soil moisture, snow water equivalent and vegetation optical depth, reanalysis radiation and air temperature, and a multi-source precipitation product. Versions 3b and 3c are satellite based with common forcing data excluding soil moisture and vegetation optical depth as these are based on different passive and active microwave sensors, i.e., ESA CCI for v3b and Soil Moisture and Ocean Salinity (SMOS) for v3c (Martens et al., 2017).

The different components of terrestrial processes (i.e., transpiration, open-water evaporation, bare soil evaporation, sublimation and water loss) are separately driven in GLEAM (Martens et al., 2017). Each grid cell in GLEAM contains fractions of four different land cover types namely: open water (e.g. dam, lake), short vegetation (i.e., grass), tall vegetation (i.e., trees) and bare soil. These fractions are based on the global vegetation continuous field product (MOD44B) with the exception of the fraction of open water. The MOD44B product is based on the moderate resolution image spectroradiometer (MODIS) observations (Martens et al., 2017). Soil moisture is estimated separately for each of these fractions and then aggregated to the scale of the pixel based on the fractional cover of each land cover type. Root zone soil moisture is calculated using a multi-layered water balance equation which uses snow melt and net precipitation as inputs, and drainage and evaporation as outputs (Miralles et al., 2011). The depth of soil moisture is a function of land-cover type comprising one layer of bare soil (0-10 cm), two layers for short vegetation (0-10, 10-100 cm) and three layers for tall vegetation (0-10, 10-100, and 100-250 cm) (Martens et al., 2017).

## 2.4    Analysis approach

### 2.4.1    Statistical analysis

The first part of the model evaluation focuses on evaluating the monthly time series data of soil moisture products at the site level. The monthly temporal scale is considered because, on very short time scales such as daily and hourly, local effects can lead to a pronounced noise in the observations. Such noise however is anticipated to be filtered through long term averaging. At a monthly time scale, the soil moisture seasonal cycle is well developed. The second part of the evaluation inter-compares model simulations and satellite estimates of soil moisture at a regional level. Time series data for the sites were extracted from the soil moisture products, using the flux tower's geographical coordinates. The satellite products present averaged soil moisture data per grid cell. A distance-weighted average (DWA) technique was used to interpolate the CCAM-CABLE model simulations to estimate soil moisture values representative of observational sites. The DWA method proved to be more representative than the nearest neighbour (NN) method, as the DWA method interpolates to the exact location of the tower by considering simulated values at grid points surrounding the location. It is noteworthy, that a comparison between the in-situ observations and satellite products in this study places much emphasis on phase agreement (e.g. seasonal cycle), as opposed to that of magnitudes. For satellite observations, GLEAM model estimates are represented as spatial averages for each pixel, in which case an interpolation of such aerial averages to a point (i.e., site), do not add further information that correspond to the site.  It may be expected that data at the point and grid box scales should still comparatively present qualitative features that are characteristic of the climate system for the region, for example seasonal cycles.

The soil moisture products were first converted to the percentage volumetric soil moisture amounts for comparison purposes. As in Yuan and Quiring (2017), we assume that the soil moisture measurements at the depth of 5 cm represent the 0–10 cm depth. In situ data at the depths of 15, 30

and 40 cm were combined using the depth weighted average method to represent the 10-100 cm depth using Eq. (1):

$$SM_{10-100} = \sum_{i=1}^{n} \frac{LT}{SD} \times SM(i) \tag{1}$$

where $SM_{10-100}$ is the weighted soil moisture, $n$ is the number of layers, $LT$ is the layer thickness calculated as the difference between the soil depths, $SD$ is the total soil depth of the soil profile and $SM(i)$ is the daily in situ soil moisture values at the i[th] layer. The depth weighted average method as presented in this study (Eq. 1) has been used in other studies such as that by Yuan and Quiring (2017). Similarly, the data at depths 2.2 and 5.8 cm, and 15.4 and 40.9 cm from CCAM-CABLE are averaged to represent 0-10 and 10-100 cm respectively using Eq. (1). Daily data from all the soil moisture products are averaged to monthly where 80 % of the daily data is available. Months that do not meet the 80 % threshold are excluded from the analysis.

**Table 1.** Overview of soil moisture datasets; satellite (grey) in percentage, modelled (blue), simulation (pink) and in situ observations (green) presented as a ratio (m$^3$ m$^{-3}$) of soil to moisture per unit area.

| Soil moisture product | Spatial resolution (km) | Spatial coverage | Soil depth (cm) | Period |
|---|---|---|---|---|
| ESA-Combined | 25 | Global | 0-10 | 1978-2015 |
| ESA-Active | 25 | Global | 0-10 | 1991-2015 |
| ESA-Passive | 25 | Global | 0-10 | 1978-2015 |
| CCAM-CABLE | 8 | Regional | 2.2, 5.8, 15.4, 40.9, 108. 5, 287.2 (bedrock) | 2000-2014 |
| Skukuza | Point data | Point | 5, 15, 30, 40 | 2000-2017 |
| Malopeni | Point data | Point | 5, 15, 30, 40 | 2008-2017 |
| GLEAM v3a | 25 | Global | 0-10, 10-100 | 1980-2016 |
| GLEAM v3b | 25 | Global | 0-10, 10-100 | 2003-2015 |
| GLEAM v3c | 25 | Global | 0-10, 10-100 | 2011-2015 |

To evaluate how similar the soil moisture simulation and model estimates are to in situ measurements, we used the stream flow plots, and the coefficient of determination ($R^2$), as defined in Koirala and Gentry (2012).

The soil moisture products used in this study (Table. 1) are under the same latitude and longitude projection. All the soil moisture projections are at the same spatial resolution of 25 km, with the exception of the CCAM-CABLE model with a resolution of 8 km. The bilinear interpolation method

was used to resample the CCAM-CABLE simulations from 8 to 25 km to match the resolution of the other soil moisture products.

The "SoilGrids" dataset from the international soil reference information centre (ISRIC) was used in this study, to map soil types (Fig. 1c). The data are available online (https://soilgrids.org), and is described in detail in the study by Hengl et al., (2017). This dataset has a spatial resolution of 250 m and was used in this study to partition the MI between simulated and modelled soil moisture according to soil type per grid box. Soil was classified into 12 dominant types ranging between sand and silty clay as described in Fig. 1d. The soil type data are available at seven depths (i.e., 0, 5, 15, 30, 60, 100 and 200 cm), here we only consider the data representing the surface (i.e., 0-5 cm). The 250 m dataset was resampled to 25 km, firstly by resampling to 1 km and then to 25 km, using the nearest neighbour method to match the resolution of the soil moisture products. We acknowledge that resampling from fine to coarse resolution might introduce bias towards certain soil types. However, the nearest neighbour method is suitable for resampling categorical data.

### 2.4.2 Cross-wavelet analysis

The cross-wavelet method analyses the frequency structure of bivariate time series using the Morlet wavelet (Veleda et al., 2012). The wavelet method is suitable for analysing periodic phenomena of time series data, especially in situations where there is potential for frequency changes over time (Rosch and Schmidbauer, 2018; Torrence and Compo, 1998). This method has been used in other studies, such as that by Koirala and Gentry (2012), for investing the climate change impacts on hydrologic response. Cross-wavelet analysis provides suitable tools to compare the frequency components of two time series, thereby drawing conclusions about their synchronicity at a given period and time. A continuous wavelet leads to a wavelet transform of a time series which preserves information of both time and frequency resolution parameters. The transform can be partitioned into imaginary ($\mathrm{Im}$) and real ($\mathrm{Re}$) parts, which provide information on both the phase and amplitude over time. This is a prerequisite in the investigation of coherency between two time series (Rosch and Schmidbauer, 2018). The Morlet wavelet is given by:

$$\psi(t) = \pi^{-1/4} e^{iwt} e^{-t^2/2} \tag{4}$$

where $w$ is the angular frequency and $t$ is time. The Morlet wavelet transform of a time series $x_t$ is then denoted by

$$Wave(T,S) = \sum_t x_t \frac{1}{\sqrt{S}} \psi^* \left( \frac{t-T}{S} \right) \tag{5}$$

where $T$ is a time parameter, $S$ is the scaling parameter and $*$ represents the complex conjugate. The cross-wavelet transforms two time series $x_t$ and $y_t$ with respective wavelet transforms $Wave.x$ and $Wave.y$, decomposes the Fourier co- and quadrature-spectra in the time frequency (or time-scale) domain. The cross-wavelet implemented is rectified according to Veleda et al. (2012):

$$Wave.xy(T,S) = \frac{1}{S} \times Wave.x(T,S) \times Wave.y^*(T,S) \tag{6}$$

Its modulus can be interpreted as a cross-wavelet power which lends itself with certain limitations to an assessment of the similarity of the two series wavelet power in the time frequency domain:

$$Power.xy(T,S) = |Wave.xy(T,S)| \qquad (7)$$

In a geometric sense, the cross-wavelet transform is comparable with the covariance. Graphically the cross-wavelet spectrum provides the cone of influence and contour lines indicating significance of joint periodicity and for checks of consistency. Information on the synchronisation of two time series in terms of phase is also presented on the plot. Phase difference of the two time series at each time scale is given by:

$$Angle(T,S) = Arg\left(Wave.xy(T,S)\right) \qquad (8)$$

equivalent to the difference of individual phases, $Phase.x - Phase.y$. When converted to an angle in the interval $[-\pi, \pi]$, this is indicated by arrows (see Fig. B1 in Appendix B) in the cross-wavelet power plot. The phase is computed using:

$$Phase(T,S) = \tan^{-1}\left(\frac{\text{Im}\left(Wave(T,S)\right)}{\text{Re}\left(Wave(T,S)\right)}\right) \qquad (9)$$

The cross-wavelet analysis described is computed in R software using the "WaveletComp" package. Detailed information on the package can be found in Rosch and Schmidbauer (2018). The cross-wavelet analysis only applies to complete datasets (i.e., without missing values). Since the in situ observations have missing data, the multiple imputations method is used to gap fill the missing parts of the in situ time series. The multiple imputations procedure is extensively discussed in studies by
Rubin (1987) and Rubin (1996) and is implemented in this study using the "Amelia" package in R. The number of imputed datasets was set to five and combined using Rubin's rules as outlined in Rubin (1996). Multiple imputations of the in situ observations are only applied to the Skukuza dataset for both the surface (Fig. C1.a) and rootzone (Fig. C1.b). This is because the Skukuza data has fewer gaps than Malopeni (Fig. A1.b). The imputed soil moisture observations are shown in Appendix C
together with the statistics of the measures of the distribution for both the gap filled and non-gap filled datasets. The cross-wavelet analysis is applied to non-stationary data using the default method (i.e., white noise) with the simulations repeated ten times.

### 2.4.3   Onset and offset of the wet period

In addition to the analysis of phase agreement, we compare the simulation of the onset and offset of
the wet periods, by the different soil moisture products and the imputed in situ observations. Instead of using precipitation as discussed in Shongwe et al. (2015) and Liebmann et al. (2007), to identify the onset and offset of the rainy season, we use soil moisture data. These are computed using a cumulative quantity over time as

$$A(day) = \sum_{n=1}^{day}\left[S(n) - \bar{S}\right] \qquad (10)$$

where $S(n)$ is the daily soil moisture and $\bar{S}$ denotes the annual daily average. As in Liebmann et al. (2007) we start the calculation on the climatologically driest month, i.e., 1 July (Shongwe et al., 2009), and perform a cumulative sum over a period amounting to a year. The onset of the wet period

is defined as the date on which the cumulative sum reaches a minimum, and the offset, as the date on which the cumulative sum reaches the maximum (Shongwe et al., 2015).

### 2.4.4 Standardised soil moisture index

The standardised soil moisture index (SSI) as contributed in Xu et al. (2018) presents an opportunity to compare spatial and temporal variations in soil moisture. The index for given day of the year can be expressed as,

$$SSI_{ij} = \frac{x_{ij} - \mu_i}{\sigma_i} \qquad (11)$$

where $x_{ij}$ is soil moisture of the i$^{th}$ day of j$^{th}$ year, where $i \in \{1 : 365/6\}$ and $j \in \{2011 : 2014\}$, $\mu_i$ is the mean and $\sigma_i$ standard deviation of the soil moisture taken on a set of data for the i$^{th}$ day as constituted by all the considered years. The index shows how the soil moisture data from the different products deviate from the long term mean. In this case the index is computed for each grid cell for the study region (Fig. 1a). The index results are either positive (i.e. wetter than the long term average) or negative (i.e. drier than the long term average). The index is computed on data from 2011 to 2014 for all the soil moisture products (i.e. CCAM-CABLE and GLEAM). We therefore calculate the number of dry days for each model over the analysis period and link the patterns to topography and hydrological zones at different regions (Fig. 1d). The different regions of equal area are selected based on the homogeneity of the topography which happen to overlap with the hydrological zones (Department of Environmental Affairs, 2013). This is done in an attempt to link land surface processes to landscapes.

### 2.4.5 Mutual information

At the grid scale, we calculated the MI between CCAM-CABLE and GLEAM products computed on the residuals of the de-trended time series. This is done to uncover to what extent different models capture similar variations at grid scale. Therefore, the MI calculation will uncover regions where there is low inter-model prediction certainly and where there is high signal correlation between the CCAM-CABLE simulations and GLEAM estimates. The MI is calculated on a de-trended and de-seasonalised (or decomposed) soil moisture time series signal. The "stl" package in R is used to de-trend the time series into its components (i.e. seasonal, trend and residual) as discussed in Cleveland et al. (1990). The MI between two variables $X$ (i.e. CCAM-CABLE) and $Y$ (i.e. GLEAM) is defined as

$$MI(X,Y) = \iint dx dy \, \mu(x,y) \log \frac{\mu(x,y)}{\mu_x(x)\mu_y(y)} \qquad (12)$$

where $\mu_x(x) = \int dy \, \mu(x,y)$ and $\mu_y(y) = \int dx \, \mu(x,y)$ are marginal densities of $X$ and $Y$ respectively (Kraskov et al., 2004). MI is estimated from the $k$-nearest neighbour statistics, in particular a simple Shannon entropy:

$$H(X) = -\int dx \, \mu(x) \log \mu(x) \qquad (13)$$

Furthermore, MI could also be obtained by estimating $H(X)$, $H(Y)$ and $H(X,Y)$ separately and using Eq. 13.

$$MI(X,Y) = H(X) + H(Y) - H(X,Y) \qquad (14)$$

The MI as outlined by equations 12 to 14 is computed in this study using the "varrank" package in R.

## 3 Results and discussion

### 3.1 Evaluation of the satellite-estimated and model-simulated seasonal cycle soil moisture

In this section we discuss how the respective outputs from CCAM-CABLE, ESA and GLEAM reflect
the key features of soil moisture for the study sites. As highlighted in the introduction section, the variability of the simulation output, satellite derived data and satellite based model estimates are studied relative to the in situ measurements from the study sites. Much focus is placed on investigating how well the seasonality of the soil moisture is reflected by respective soil moisture data sets. This is because the ability of models to capture the seasonality of a system is more important
than its agreement with observations in absolute values (Fang et al., 2016). It has also been mentioned in the introductory section that the in situ observations are taken from semi-arid savanna sites within the Kruger National Park. The patterns of soil moisture at these sites are mainly driven by rainfall which is generally higher during the summer season, and low in winter as shown in Fig. 2. The long term surface soil moisture for both of the sites follows a pattern comparable to that of rainfall as can
be seen by comparing the soil moisture patterns presented in Fig. 2 (i.e. long term cycles) and the monthly rainfall accumulations (Fig. 2).

### 3.1.1 Long term seasonal cycles

In general the pattern of the long-term average for soil moisture (Fig. 2) from the CCAM-CABLE simulations, ESA satellite observations and GLEAM model estimates are qualitatively comparable to
425 that of in situ observations. Notably, the observed soil moisture seasonal cycle at the surface at both Skukuza and Malopeni surface displays a local maximum in April and shows and increases from September to December and January. Soil moisture amplitudes are less pronounced in the root zone, but with November and October maxima at Skukuza and Malopeni, respectively. These patterns are consistent with the observed rainfall cycle of cessation of the rainy season in April and with onset in
October. The root-level soil-moisture pattern displays the signatures of soil moisture retention, which relates to the persistence of dry and wet periods at various soil depths (Seneviratne et al., 2006). In light of this, it is also interesting to see how both the CCAM-CABLE simulation and the GLEAM products depict the onset and cessation of the rainy season, an aspect to be discussed further in Section 3.2. The CCAM-CABLE model simulates soil-moisture to peak in March rather than April
for Skukuza at 0-10 cm, and it also does not simulate the maximum recorded soil moisture for Malopeni in April at the surface. This is probably due to the fact that the CABLE soil-moisture scheme does not take into account soil resistance (Whitley et al., 2016). Despite this, the long term CCAM-CABLE monthly means of soil moisture are relatively comparable to in situ observation even in terms of magnitude (Fig. 2). GLEAM v3c, on the other hand, agrees with in situ measurements on
the existence of an April soil moisture maximum, but it reflects the point-observed November increase in soil moisture a month earlier (i.e., in October). The satellite products (i.e., the active, passive and combined ESA products) and GLEAM models (Fig. 2) display the same signal as that of

the observed soil moisture, indicating that the April maximum in particular is not an artefact of the point observations. We can safely deduct that the bias in GLEAM v3c is not induced by satellite-based forcing data. However, this calls for further investigations on the sensitivity of the model to its driving data at a high resolution. We anticipate that at high temporal resolution there is a strong variability in the in situ soil moisture signal which may not entirely be captured, by both CCAM-CABLE and GLEAM, possibly due to their relatively low spatial resolution. The relatively low resolution (8 km in the horizontal) in the case of CCAM-CABLE, in particular, potentially has strong implications for how representative the effective drivers of soil moisture such as soil texture and vegetation covers are in terms of observations at specific sites.

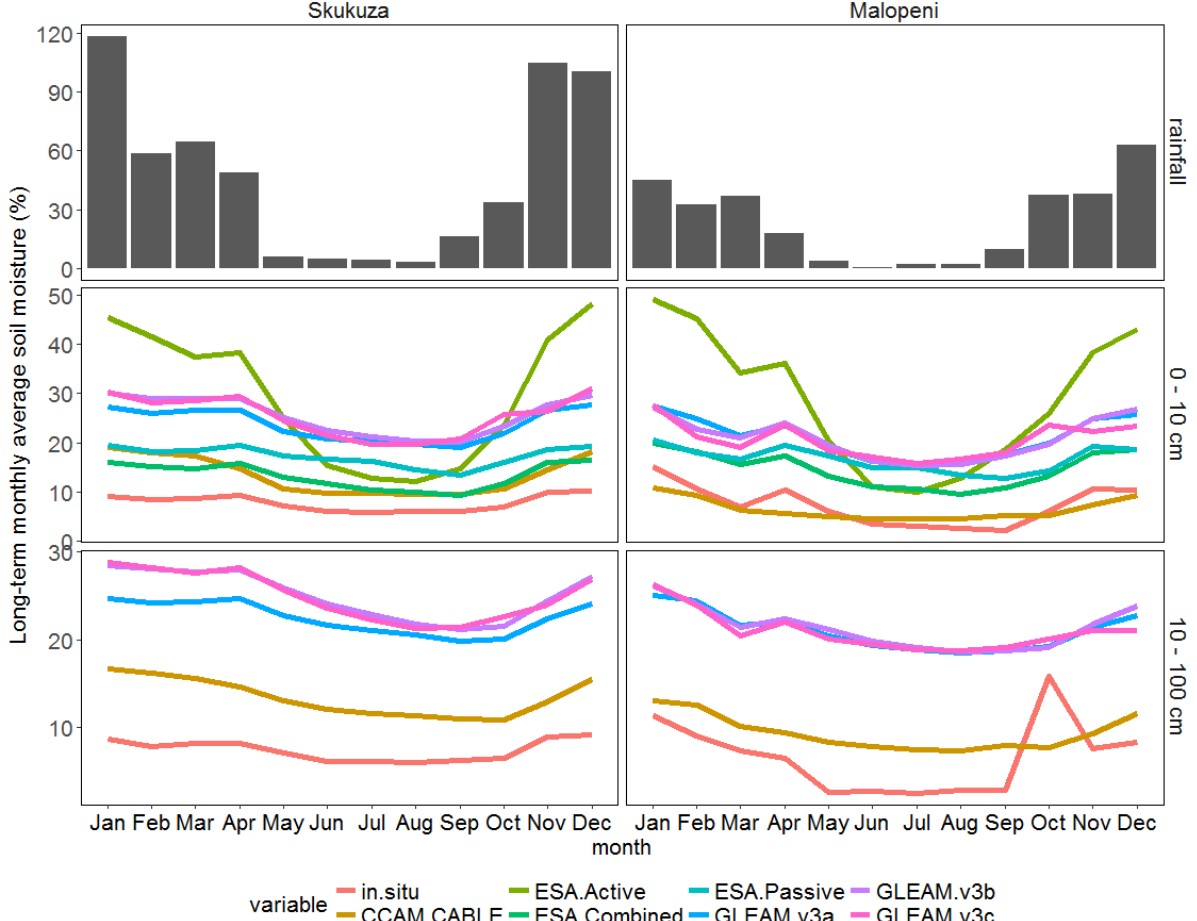

**Figure 2.** Seasonal variation in the long term mean monthly rainfall (mm), surface (i.e., 0-10 cm) and root zone (i.e., 10-100 cm) soil moisture, based on in situ observations and a variety of soil moisture products. The in situ data is collected from two sites, namely Skukuza (2001-2014) and Malopeni (2008-2013).

Soil moisture is at its lowest during the dry periods (i.e., May to September) and highest during the wet periods (i.e., November to April). The GLEAM models (Fig. 2) are generally consistent with in situ measurements in estimating soil moisture both in terms of magnitude and phase, both at the surface and root zone. The magnitude of GLEAM v3a root zone estimates is lower than those of the other GLEAM models at the Skukuza site. This can be attributed to the unique multi-source weighted ensemble precipitation (MSWEP) data used to force GLEAM v3a (Martens et al., 2017), which is different to the precipitation forcing data used in GLEAM v3b and GLEAM v3c. We further observe that the GLEAM models, satellite and in situ observations have the same length of the dry period, with the exception of the ESA-Active observation which has a shorter dry period. The ESA-Active

satellite product is known to work best for moderate to densely vegetated areas as opposed to savanna sites such as Skukuza and Malopeni where tree cover is sparse (Dorigo et al., 2015) and the vegetation cover changes dynamically due to a combination of factors, for example fires and rainfall. There is less difference between the ESA-Passive and ESA-Combined satellite products. Generally, the ESA-Combined and ESA-Passive datasets have the least difference during the dry period for all sites. The ESA-Combined product shows a strong increase in soil moisture in July which is not observed in the other soil moisture products. This increase in the ESA-Combined product may be due to irrigation detected by satellites which is not accounted for by the models (Al-Yaari et al., 2019). Using long term averages, both the CCAM-CABLE and GLEAM models are able to capture the intrinsic seasonality of the soil moisture signal for the sites as reflected by both in situ and satellite observations. This is despite their being different both in the forcing data and model structure. Studies by Wang and Franz, (2017) and Seneviratne et al., (2010) suggest that local factors (e.g., vegetation, soil and topography) mostly control soil moisture variability at spatial scales less than 20 km, rather than meteorological forcing. For a fourteen-year averaging period, undoubtedly the monthly means are sensitive to anomalously high precipitation, and hence soil moisture, in some months. It is therefore instructive to investigate how well the simulated and estimated patterns of soil moisture compare with the in situ data on a monthly basis for the respective years.

### 3.1.2  Inter-annual variability in the seasonal cycle

This section presents quantitative evaluation of the soil moisture time-series from the CCAM-CABLE simulations and GLEAM estimates at a monthly time-resolution for the 2001-2014 period. To circumvent possible bias due to missing values, monthly averages are presented only for months where there are observations above the imposed 80% data availability threshold. This implies that the number of data points (i.e., sample size), are not equal for all products. The different model, satellite and point data sets are also not available for equal time periods. For example, GLEAM v3c has the shortest data set spanning between 2011 and 2014, as opposed to the point observations that range between 2001 and 2014 for Skukuza, and 2008 and 2013 for Malopeni. The monthly time series data as well as data availability for the different soil moisture products are presented, and qualitatively compared with in situ observations in Fig. 3 in Appendix A for both the Skukuza and Malopeni sites. Generally, we observe in Fig. 3 agreement in the simulation of short term seasonal cycles as compared to in situ observations especially at the surface. This is reflected for most of the years, and indicates that the observed seasonal cycle is present in the simulated and satellite estimated soil-moisture time-series. At the root zone, we observe a decrease in the similarities of the cycles between the simulation, satellite-estimates and in situ observations. Clearly this requires further investigation into water drainage and soil moisture memory which is outside the scope of the discussion in this study. We further observe that for these cycles the soil moisture products compare best to in situ data, than they do to the CCAM-CABLE simulations as shown in Fig. A1.

The similarities between these short term seasonal cycles are quantified in Fig. 3 using $R^2$ values. On account of missing values, the $R^2$ values presented in Fig. 3 are based on different sample sizes, therefore, their interpretation is made with this issue in mind. The $R^2$ values are generally higher at Malopeni compared to Skukuza, indicating that the few months where there are observations, there is also a high comparability of the signal. It is however, inconclusive whether the simulations and estimates are more comparable at Malopeni relative to the case in Skukuza. In general, based on Fig. 3 all the soil moisture products are able to capture the variability in the observed soil moisture, mainly exceeding the $R^2$ value of 0.5 (i.e. 50%) both at the surface and root zone. The CCAM-CABLE model mainly presents the least, but reasonable $R^2$ value, both at the surface and root zone. This is a further

510    reflection that the simulated soil moisture cycle is stronger in amplitude compared to in situ observations. Overall, the relatively large $R^2$ values displayed in Fig. 2 stem from the model and satellite products ability to realistically represent the seasonal cycle in soil-moisture.

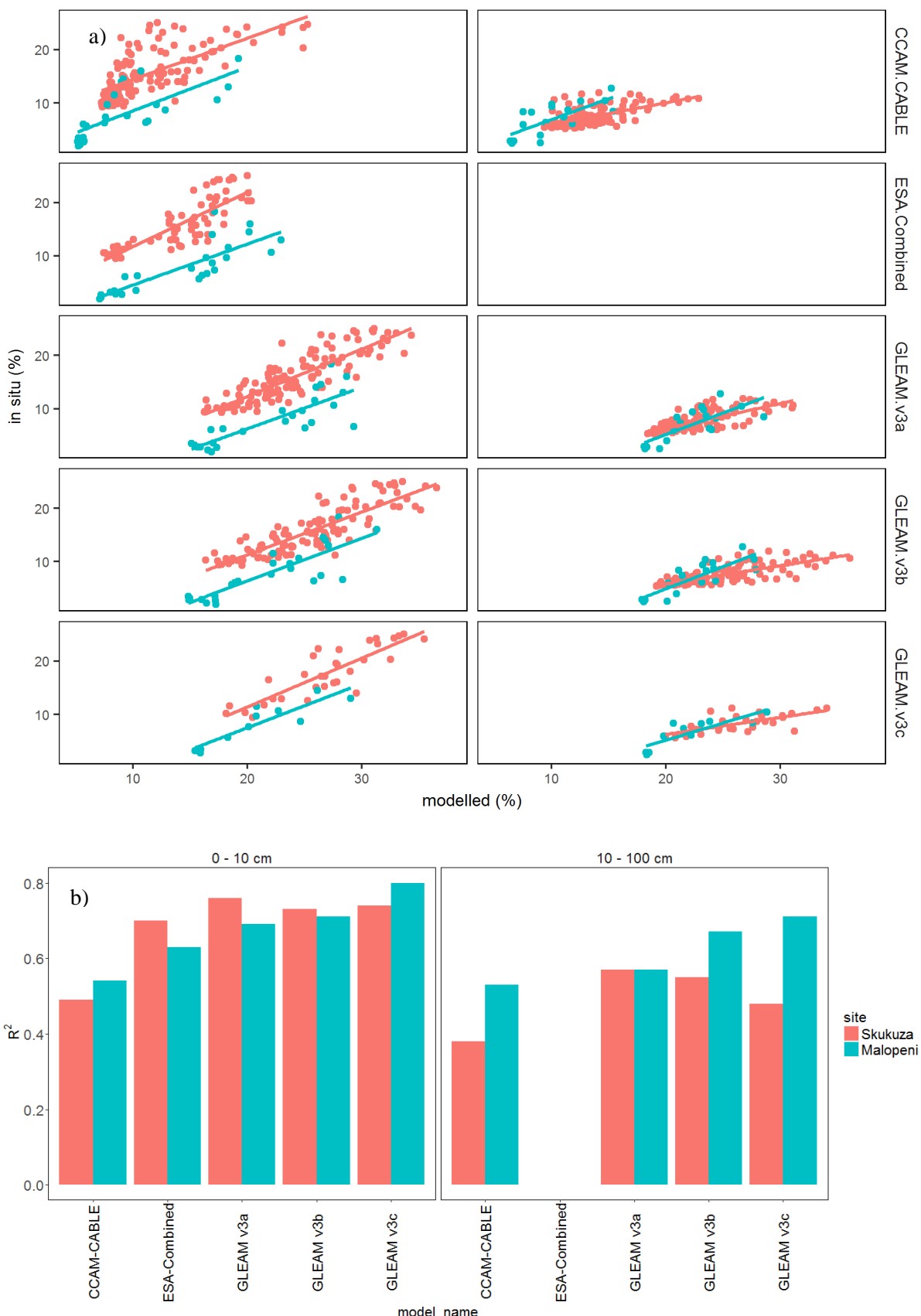

Figure 3. a) Quantitative comparison between soil moisture products and observations at Skukuza (red bars) and Malopeni (green bars), at the surface (0-10 cm) and root zone (10-100 cm), using the b) coefficient of determination ($R^2$),

The ESA-combined satellite product is expected to present the best agreement with observations, since it is actual observed data from the sites. This expectation is realised with $R^2$ values greater than 0.65. Furthermore, the ESA data has been shown to generally capture soil moisture in different regions and climate zones of the world (Loew et al., 2013; McNally et al., 2016; Wang et al., 2016; Zeng et al., 2015). Our study confirmed (Fig. 3) that the ESA combined product captures local conditions within reason/acceptable amount of certainty. A study conducted by Yuan and Quiring (2017), assessing the performance of CMIP5 models, both at the surface and root zone, concluded that the models performed better at the root zone, compared to the surface. This is contrary to our findings in this study, where we generally observe improved agreement between soil moisture products and in situ measurements at the surface than at the root zone. Based on the extent to which GLEAM products proved to be representative of the qualitative features of the soil moisture signal for different months and seasons, as driven by precipitation at the site, it is compelling to further resolve qualitatively how the simulated output compare with each other for most of the time periods. To this effect, we next present the results from a cross-wavelet analysis of CCAM-CABLE simulation output and GLEAM estimates for the two study sites.

### 3.1.3  Cross-wavelet analysis

The cross-wavelet power spectrum (Fig. 4) reveals that, generally the soil moisture time series of CCAM-CABLE simulations and GLEAM v3a are in phase with the in situ observations time series at the Skukuza site at the soil depths investigated. This is indicated by the arrows generally pointing to the right, as demonstrated in Fig. B1 in appendix B. The arrows are plotted between the white contour lines indicating areas of significance, and joint periodicity at 10 % (i.e., 90 % confidence level). The cross-wavelet power spectrum (Fig. 4) shows the strength of the variation between the two signals as function of frequency. The red colours indicate weak variation (i.e., strong signal or agreement), while the blue colour indicate strong variation synonymous to random noise. This area of significance is generally between the periods of 8 and 15 months (y-axis). Although the time series are in phase most of the time, in some instances there is a lag. This is identified by the direction of the arrows, in which case the arrows are inclined upwards or downwards at different margins. For example, there is a lag of two days on average between CCAM-CABLE simulations and in situ observations at the period of about 12 months, and a lag of about six days on average between GLEAM v3a and in situ observations at the surface at Skukuza. At the root zone we observe a wider lag between observations and the soil moisture products, there is a lag of 14 and 24 between the soil moisture products (i.e. CCAM-CABLE and GLEAM-v3a) and in situ observations respectively. This implies that there is better agreement between the soil moisture products and in situ observation at the surface than at the root zone. A plot demonstrating the phase differences, between in situ observations (red) and CCAM-CABLE (blue) extracted in Fig. 6a, for the years 2001 to 2014 associated with soil moisture patterns with a characteristic period of 12 months, is illustrated in Fig. B2 in appendix B. Fig. B2 show that there is an exchange of leading and lagging of the two time series. For example, the in situ observations lead CCAM-CABLE at the beginning of the two time series, the two time series are in perfect phase in the middle and the towards the end CCAM-CABLE leads and the in situ observations lag. When there is no lag between the two series, particularly for the annual cycles it indicates the repeating features of the respective signals which repeat between 11 and 13 months for the surface.

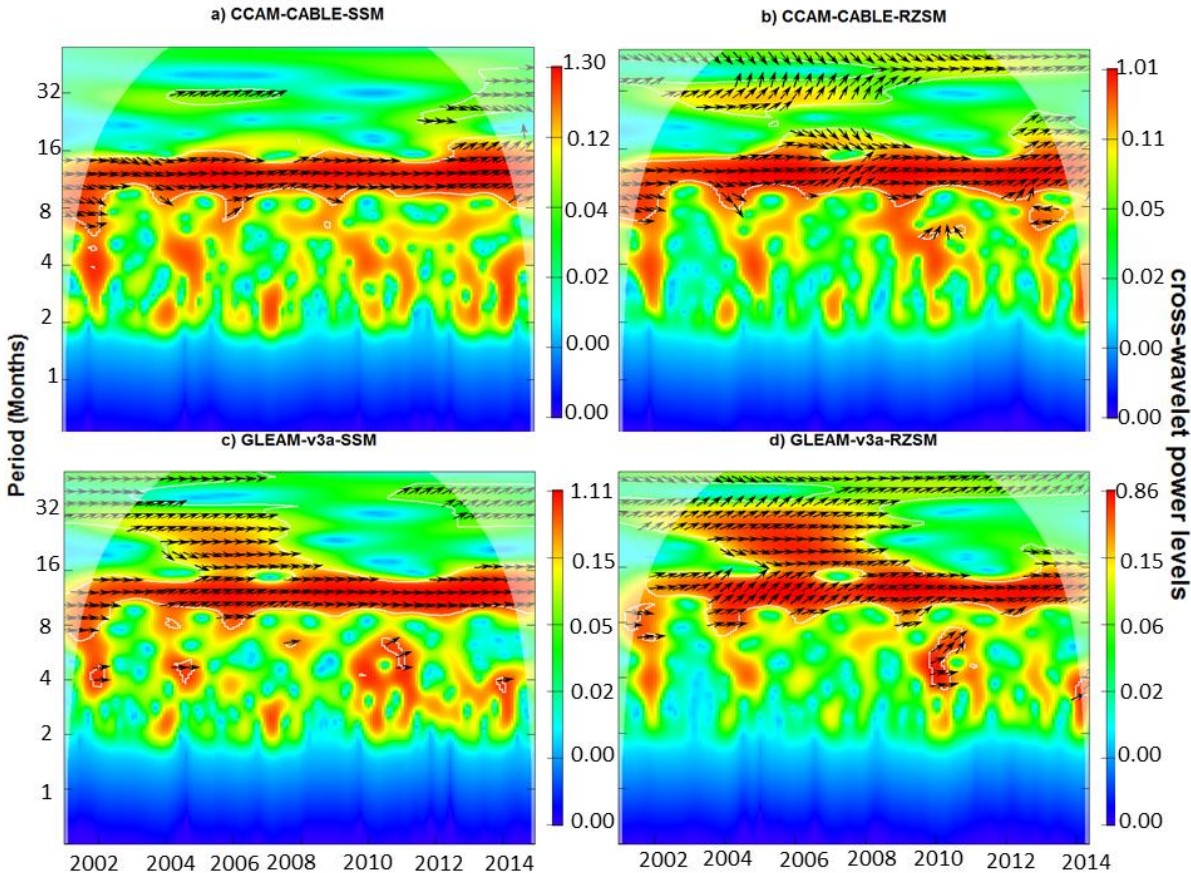

**Figure 4.** Cross wavelet power spectrum of surface (SSM, 0-10 cm) and root zone (RZSM, 10-100 cm) soil moisture between in situ observations, CCAM-CABLE (a, b) and GLEAM v3a (c, d) at Skukuza respectively. The white contour lines indicate periods of significance at 10 %. The arrows pointing to the right indicates that the models are in phase, anti-phase point left, in situ observations leading either CCAM-CABLE or GLEAM v3a is indicated by arrows pointing straight down. The dome shape (shaded areas) represents the cone of influence between 2001 and 2014. The red colour indicates weak variation while blue indicates strong variation.

Furthermore, we employ the cross wavelet between CCAM-CABLE and GLEAM-v3a at the two study sites in Fig. B2, in Appendix B., The results shows that between the year 2002 and 2005 at periods of about 12 months, we observe that GLEAM v3a lead and CCAM-CABLE lags by 6 days on average at the surface at Skukuza (Fig. B2.a), and by 10 days at Malopeni (Fig. B2.c). The cross correlation also shows that, there are other cyclical responses of the soil moisture signal with a periodicity of approximately two years. This becomes apparent for some years when the CCAM-CABLE and GLEAM v3a signal have statistically significant periodic features which repeat after 28 months. In this case CCAM-CABLE leads GLEAM v3a by 5 days on average. At the root zone we see a similar pattern as that of the surface soil moisture. The most statistically significant shared periodic features between in situ observations, CCAM-CABLE and GLEAM v3a have periods mainly between 10 and 16 months. This is true for the entire time series (i.e., 2001-2014). The cross-wavelet analysis in this case picks the characteristic annual pattern of soil moisture which is effectively repeated for different years. The time series are in phase for the whole analysis period generally without any lag between 2007 and 2012 for periods ranging between 9 and 15 months. For the feature of the signal with a 12 month period, there is on average a time lag of 10 and 19 days at Skukuza (Fig. B2.b) and Malopeni (B2.d) respectively. We further note that the significant periodic features of the signal generally increase from the surface to root zone. This is potentially associated with differences in the drivers of soil moisture between the respective layers. Root zone soil moisture for instance, is

likely to respond to plants driven moisture demands in a slightly different manner in comparison to
585 the surface layer. An accurate attribution of soil moisture patterns per layer to the respective drivers,
in this context, is a rather complex problem and demands a separate investigation.

The simulation and the models estimates show coherency in capturing periodic patterns, at least those
that are recurrent on an annual time scale. The cross-wavelet analysis successfully reveal that there is
a similarity in the patterns of surface and root zone soil moisture over time at both sites albeit
negligible differences across-sites for events that are recurrent on periods below or exceeding 12
months. The existence of a time lag or differences in phase in the soil moisture signal between the
simulation and GLEAM model outputs, is likely a result of the non-deterministic nature of
thunderstorm formation in the CCAM-CABLE simulations, despite these runs being nudged within
ERA reanalysis data (see Section 3.1.2). The simulations are nevertheless expected to capture most of
595 the characteristic features of the climatic system such as seasonality. Its output may not match satellite
derived observations on certain aspects including day-to-day variability. Some correspondence is
expected between the CCAM-CABLE simulations and the GLEAM outputs in terms of inter-annual
variability (Dedekind et al., 2016).

Next, we explore how these differences reflect on the onset and offset of the wet period calculated
from the in situ observations, CCAM-CABLE simulation and GLEAM estimates. The results in Fig. 2
indicate that the modelled and satellite derived soil moisture products generally capture the length of
both the dry and the wet period. However, Fig. 4 shows that there is a lag between the time series of
in situ observations, CCAM-CABLE and GLEAM estimates which indicate uncertainty in phase
agreement. We generally observe in Fig. 5 that there is agreement in the estimation of the onset and
605 cessation of the wet period amongst the different soil moisture products. The onset (Fig. 5, triangles)
of the wet period is generally between September and December, as shown by the different soil
moisture products at the two sites. Cessation (Fig. 5, circles) is generally between January and May of
the next year following the onset. We observe agreement in the occurrence of the onset and cessation
of the wet period in the in situ observations, CCAM-CABLE simulations and GLEAM estimates in
some years. Figure 5 show that the GLEAM models have a relatively low uncertainly for the onset
and cessation of the wet period. This is expected as these models use similar forcing data. For
example, GLEAM v3a and v3b agrees on the onset of the wet period during the following years;
2001, 2007, 2008, 2009 and 2013 at Skukuza. Furthermore, GLEAM v3a and v3b agrees on the offset
of the wet period during 2007, 2008 and 2011. The CCAM-CABLE and GLEAM products
predominantly differ by a factor not exceeding 30 days on the timing of the onset of the wet period.
There is a very noticeable uncertainty in the timing of the cessation of the wet period among all
approaches. These analysis yield results that are consistent with those observed in Fig. 4. The study
sites mainly experience summer rainfall, commonly occurring between November and April. The
CCAM-CABLE model generally shows a consistent length of the wet period at both study sites for
most of the years. The GLEAM models generally present an early onset in October and offset in May.
This result is consistent with the difference in phase between CCAM-CABLE and GLEAM observed
in Fig. 4.

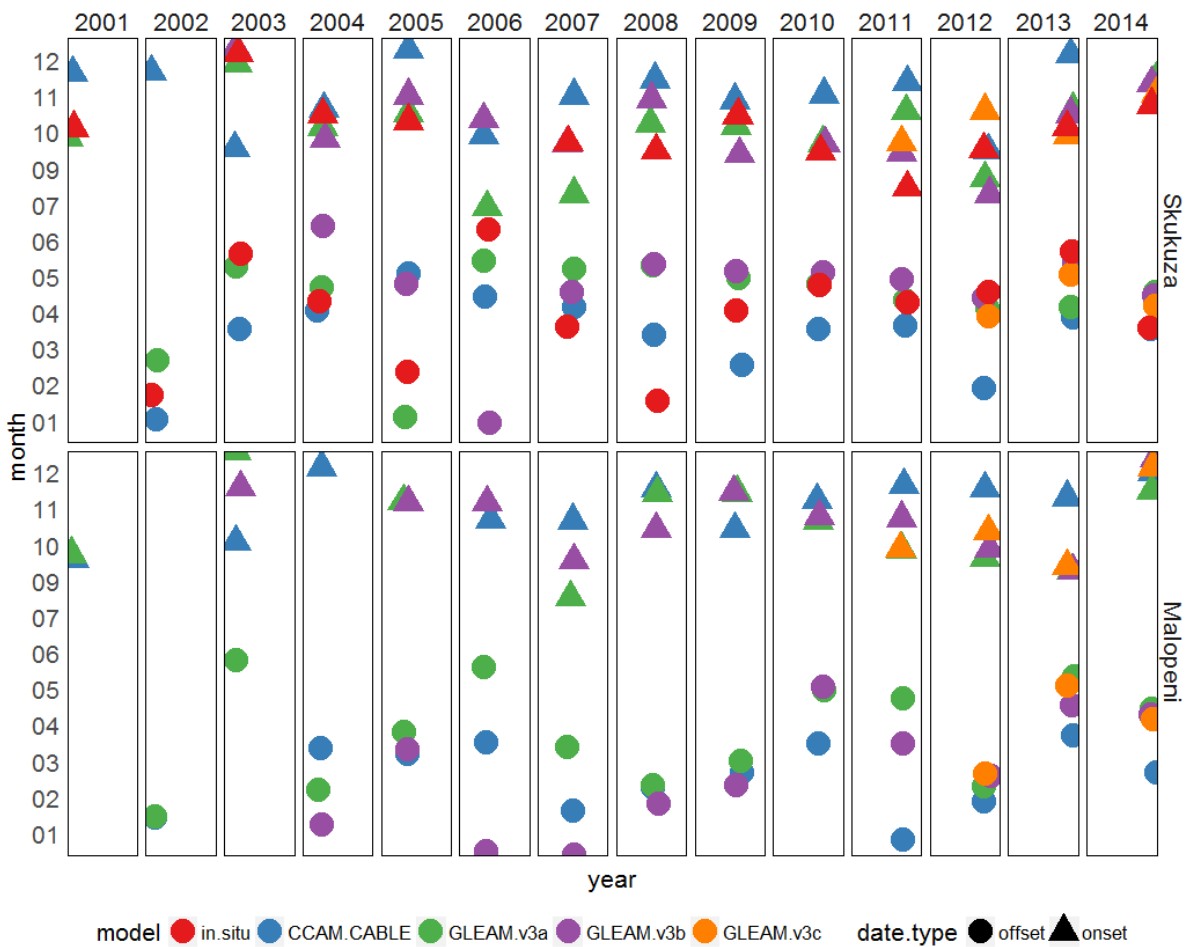

**Figure 5.** Onsets (triangles) and offsets (circles) of the wet period at Skukuza of in situ observations (2001-2014, red) and simulated by; CCAM-CABLE (2001-2014, blue), GLEAM-v3a (2001-2014, green), GLEAM-v3b (2003-2014, purple) and GLEAM-v3c (2011-2014, orange). The Malopeni plot does not show in situ observations as imputation of the data is not possible due to large gaps.

Looking at the agreement between in situ observations, CCAM-CABLE simulation output and the GLEAM model estimates, when it comes to the main periodicity of the soil moisture signal portrayed in Fig. 4 as well as the results of the onset and cessation of the wet period Fig. 5, we find that the two modelling approaches are representative of the key features of the soil moisture signal. It is also interesting to note that the level of uncertainty between the two modelling approaches, as reflected by the onset of the wet period in Fig. 5, is within an acceptable level i.e., it predominantly lies within days not exceeding a month. The uncertainty is more pronounced when it comes to the cessation of the wet period. This is indicative of differences in inter-annual variation of the soil moisture signal which is expected, to a certain extent, due to the different input data used and the mathematical structure of the models. Clearly there is need for an understanding of how the noted uncertainty could be attributed to various factors from forcing data or soil moisture drivers. It would be very important to understand, in particular, how much uncertainty is inherent in the individual coupled model key components. An in-depth investigation of various sources of model uncertainty is indeed a topical issue (Fang et al., 2016) which deserves more attention, but such a discussion will not be dealt with in this study. It is interesting to establish whether the insight gained in understanding the level of inter-comparability of the soil moisture signal, at the two respective sites, will hold at the regional level, i.e., we want to know if the mutual agreement between simulation and model estimates persists for the

study region indicated in Fig. 1b. A natural starting point is to look at the MI between the simulated and estimated soil moisture signal for the region.

## 3.2   Regional inter-comparison

In the absence of dense in situ soil moisture data providing spatial coverage over the study region, for the neighbourhood of the study sites, it will only suffice to reflect the extent to which two independent
approaches for computing soil moisture have shared information. In particular, we want to uncover how the shared information varies spatially across different landscapes such as topography (i.e. elevation), soil and vegetation types. The standardised soil moisture index (SSI) is used to uncover such patterns, and it is computed on a daily time scale annually for each soil moisture product (i.e. CCAM-CABLE and GLEAM) for the period 2011 to 2014. The index presents positive and negative
values, indicating that the index is either above or below the historic soil moisture mean for the grid cells. The index values indicate whether each pixel has above (i.e. wetter) or below (i.e. drier) soil moisture as compared to the long term mean respectively for the analysis period. The SSI as presented spatially in Fig. 6 shows the inter-annual variability of the frequency of dry days per grid cell for the study region across the models. The frequency of dry days was computed on each grid cell based on
the number of days when the SSI values were negative per year. Generally we observe in Fig. 6 consistent patterns amongst the models spatially.

The SSI permits quantification of soil moisture anomalies at various time frames and can be analysed to unearth soil moisture pattern including extremes. In this section we present the frequency count of negative anomalies of soil moisture over an annual time frame. The results indicate a pronounced
variability of soil moisture across the years. In particular, majority of grid cells had below 150 dry days during the year 2011 while for the year 2012 most grid cells are reflected as having a relatively elevated count of drier days ranging between 150 and 200 days. The pronounced count of drier days persists for the years 2013 and 2014 over the Central, South-Eastern and North-Eastern parts of the study region. In general the count of negative SSI portrays the existence of spatially connected
regions of a similar annual pattern. It is therefore a natural question to ask whether majority of grid cells belonging to a uniform elevation range could have a distinct annual SSI pattern thus reflecting a possibility of an existence of a set of common drivers of soil moisture for neighbouring grid cells at a common elevation.

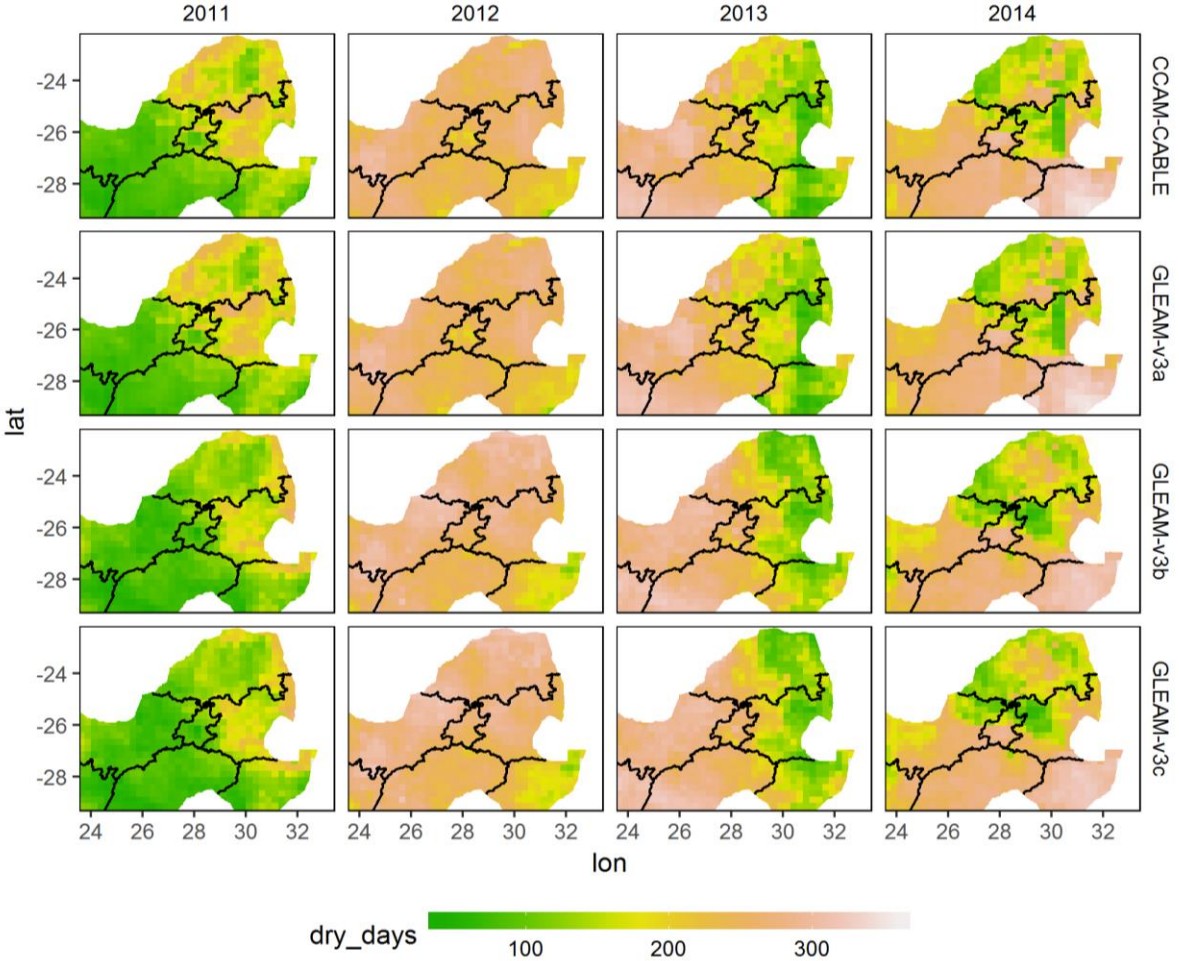

**Figure 6.** Spatial distribution of the number of dry day per year computed on the number of the negative values of standardised soil moisture index for each grid cell and model.

Next we investigate how the SSI values vary across the topographical gradient, by looking at six sub-regions (Fig. 1d) of equal size (i.e., same number of grid cells) for the years 2011 to 2014. The grid cells for the selected sub-regions belong to homogenous altitude. The sub-regions are named in the order of increasing elevation from (a)-(f). The results for this analysis are summarised through violin plots in Fig. 7. As previously mentioned, a negative SSI anomaly value indicates a dry day, while a positive anomaly value reflects a wet day, relative to the normal soil moisture value for the day. The distribution of the SSI values per selected sub-region is portrayed with a violin plot. The thicker region of each violin plot reflects the SSI value around which majority of grid cells are centred, while the height indicates the spatial spread of the SSI values. It is evident that the year 2011 had predominantly high soil moisture across most of the sub-regions albeit some low laying sub-regions (a) and (b) which had a bi-modal SSI distribution showing a predominance of SSI values lying on the wet and dry extremes. This is consistent amongst all the models. The SSI values during the year 2012 and 2014 are predominantly negative indicating a tendency toward drying relative to the long-term mean. The pattern is comparably captured by all models. During the dry years, the median of the distribution of SSI values for each of the sub-regions across the elevation gradient vary from year to year. This potentially suggests that soil moisture's key drivers towards dryness alternate over time. In particular, the temporal variability of the index may reflect that soil moisture responds differently across the elevation gradient, to the inter-annual variability in the factors influencing the climatic system.

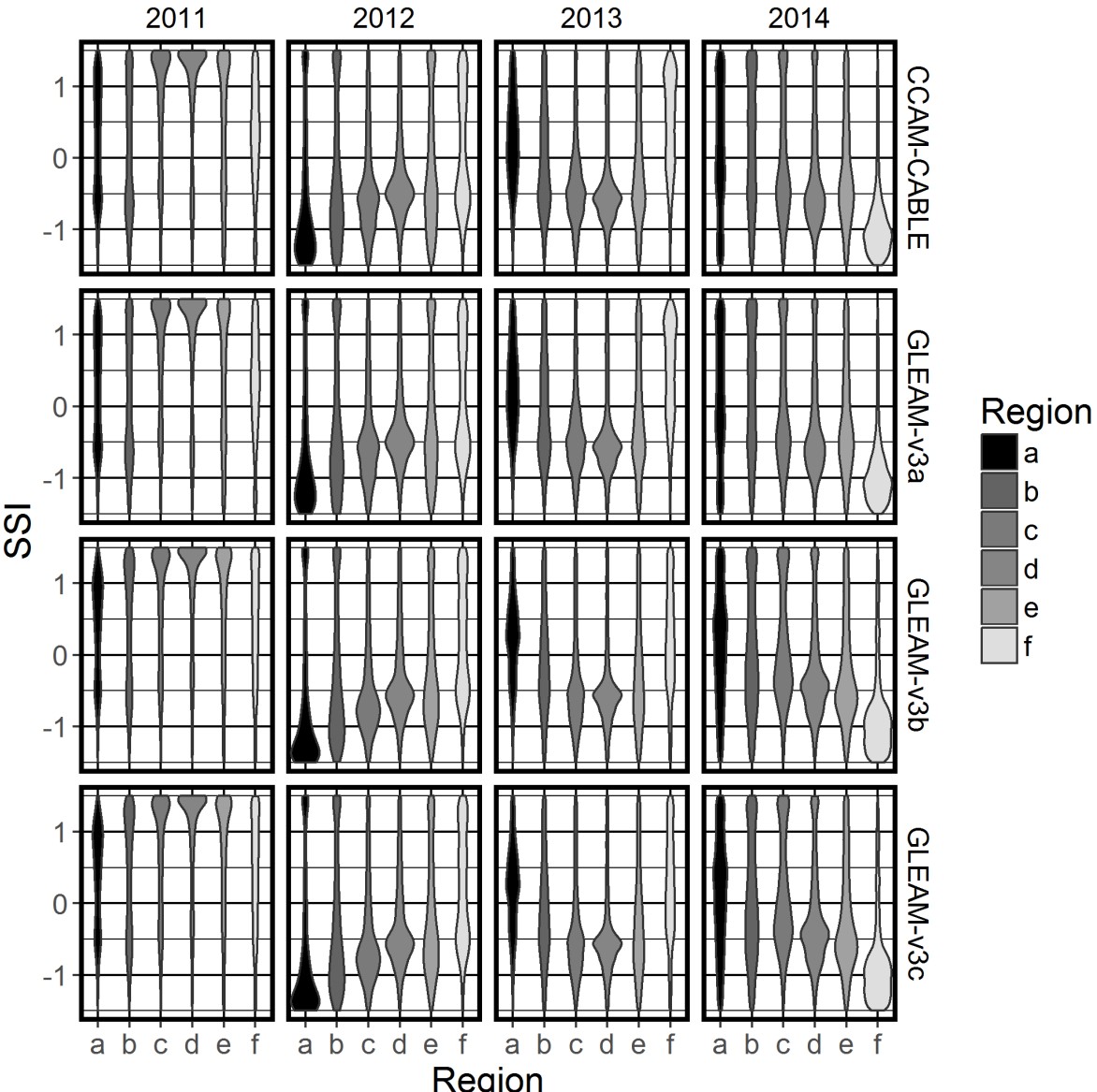

**Figure 7**. Violin plots showing the distribution of the standardised soil moisture index (SSI) values per year at different sub-regions. The regions are arranged based on increasing altitude (i.e. a-f).

Whereas the Fig. 6 and Fig. 7 indicate that that the soil moisture spatial patterns are qualitatively similar across temporal and spatial scales, the extent to which corresponding grid cells across models compare can best be quantified by a calculation of MI between the respective models. The quantification of MI is displayed in Fig. 8.

In Fig. 8, a plot of MI between soil moisture from CCAM-CABLE simulations outputs and GLEAM
model estimates is portrayed. The MI is computed from the residuals of the de-trended series of both CCAM-CABLE and GLEAM models. For this analysis, only data from 2011 to 2014 were used, as it is the common period between all the soil moisture products. We generally observe (in Fig. 8) that at the surface (SMsurf, 0-10 cm), the MI between the soil moisture simulated by CCAM-CABLE and GLEAM is higher, compared to at the root zone (RZSM, 10-100 cm), implying that shared
information between GLEAM and CCAM-CABLE is predominantly more pronounced at the surface

compared to the root zone. In particular, we observe in Fig. 8 that there is high MI between CCAM-CABLE and GLEAM-v3a at the surface compared to the rest of the GLEAM products for all the grid cells. This signals differences in the representation of soil moisture drainage at the root zone between the simulation and satellite data based model estimates. In order to see if there are any major differences in the simulation and GLEAM models estimates that can be associated with differences in soil types. We further partition the MI between the models at various soil types (Fig. 1b) of the grid. The results are presented in Fig. 9.

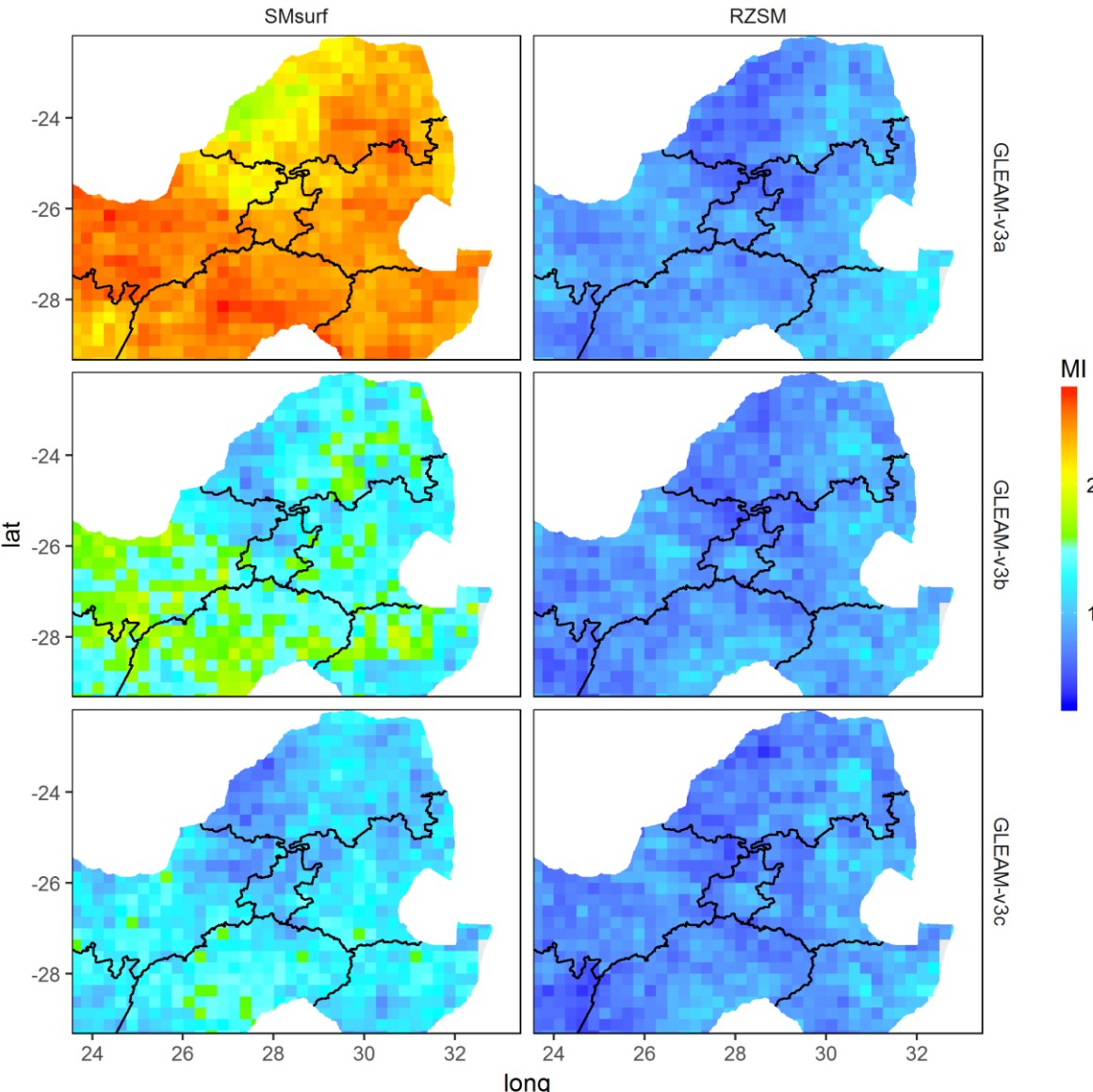

**Figure 8.** Mutual information (MI) computed on the residuals of monthly time series (2011-2014) of surface (SMsurf, 0-10 cm) and root zone (RZSM, 10-100 cm) soil moisture, between CCAM-CABLE simulations and GLEAM models estimates.

The dominant soil types in the region include loam, silt and clay (Fig. 1b). The dominant soil types 5 (loam) and 6 (silt loam); and soil types 8 (silty clay loam) and 9 (clay) as presented in Fig. 1b are associated with the grassland and savanna biomes respectively. A study by Stevens et al. (2015), the grassland and savanna biomes are dominant in this study area.

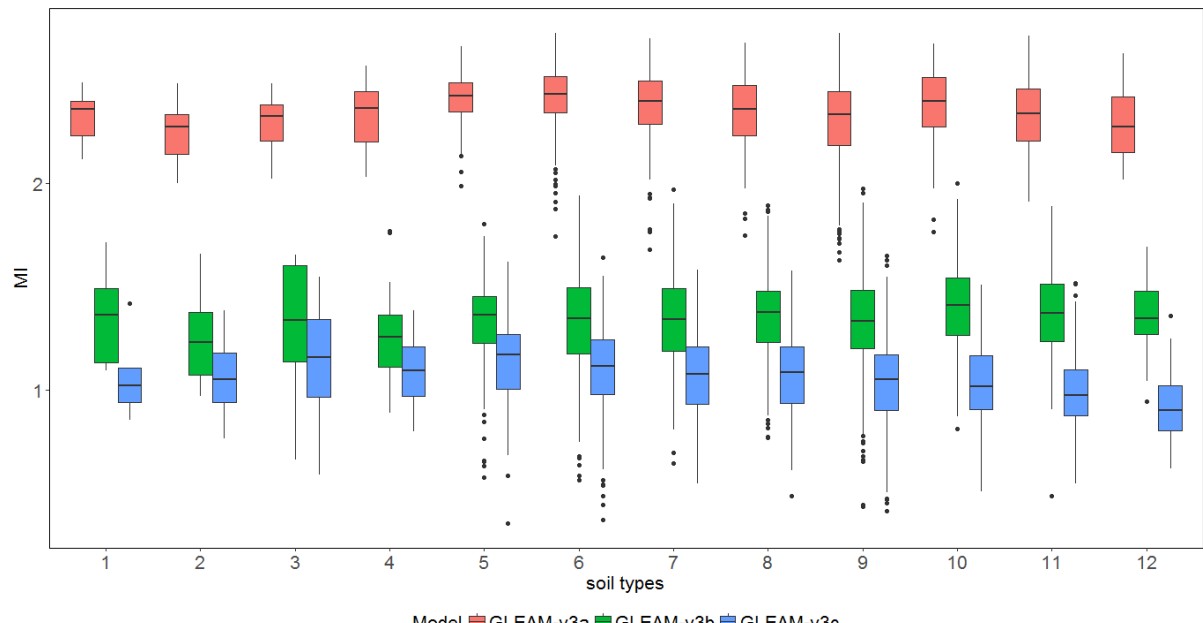

**Figure 9.** Boxplots showing the spread of the mutual information (MI) per model and soil type, the horizontal lines in the box plots represent the median. The MI of soil moisture, per soil type computed between the residuals of CCAM-CABLE and the GLEAM-v3a (red), GLEAM-v3b (green) and GLEAM-v3c (blue) soil moisture products,

The spread of the MI, as grouped by soil types, is presented in Fig. 9. Generally, we observe that GLEAM-v3a has the highest MI with the CCAM-CABLE simulations at the surface across soil types followed by GLEAM-v3b and v3c respectively, this pattern is also observed in Fig. 8. The differences in the MI across the GLEAM models are attributed to the forcing data of the individual versions. The performance of GLEAM v3b and 3c is similar as they are forced with similar data, as opposed to forcing data used in GLEAM v3a (Martens et al., 2017; Miralles et al., 2011). The MI between the residuals of CCAM-CABLE and GLEAM models is positive. In particular, the spread of the soil moisture MI generally ranges between 0 and 2.5. This is indicative that there exists joint variation even on time scales shorter than a season between the respective signals. Based on the inter-quartile range (i.e., height of the bars of the boxplots), we observe that there is a pronounced variability in soil moisture MI between CCAM-CABLE and all the GLEAM products across different grid cells. However, across the soil types the spread is mostly comparable between GLEAM-v3b and v3c. The comparability is in the sense that all box-and-whisker plots have appreciable overlaps within the inter-quartile range especially between GLEAM-v3b and v3c. This indicates comparable spatial uncertainty on the MI among soil types and models. Clearly, the resampling of the soils data done for the soil classes introduces a certain level of uncertainty. It is therefore instructive to look at the patterns central tendency as reflected by the median of the MI. The median values of the MI is a preferable measure as it is less sensitive to outliers (i.e., values beyond the whiskers ends), compared to the mean per soil category.

It is worth noting that the median MI between CCAM-CABLE and all the GLEAM models is higher at the dominant soil types (i.e. 4-sandy clay; 5-loam; 6-silt loam and 7-silt). This is indicative that there is a fair amount of data points that lie further apart from their associated grid cell mean. This could mean that the respective distributions, for a specific grid cell whose MI is calculated, have comparable synchronous points that lie apart from the mean or one of the soil type distributions is having such outlying points. The latter is likely to be predominant in the case where there are some time lags as demonstrated in Fig. 4. This alludes to differences in the representation of inter-annual

variation by the simulation and GLEAM model estimates as highlighted in section 3.1. This indicates that despite the fact that there is joint variation between the simulation and GLEAM models, there exists non-negligible variability within these respective soils types which can potentially be uncovered by studying responses to various soil moisture drivers as modelled through the respective approaches. As mentioned earlier, these dominant soil types are generally important for agricultural purposes which make it very relevant for further investigations. The least pronounced soil class MI mean between the simulation and GLEAM models occurs in soil type 12 (silty clay). This is one of the least dominant types. In this case there are very few grid-points for a meaningful comparison against other categories. In summary, from the regional MI calculation we observe that there exists a joint variation at short time scales between the simulation and GLEAM estimates. Apart from this, there is a fairly modest level of model uncertainty between CCAM-CABLE and GLEAM products, which is comparably reflected across all soil types. Attribution of its inherent sources merits further investigation.

## 4   Conclusions

In this study, the ability of a process based simulation model (CCAM-CABLE), satellite data driven model estimates (GLEAM) and satellite observations (ESA-Active, -Passive and -Combined) are evaluated against site specific in situ observations from two flux tower sites namely, Skukuza and Malopeni. The sites are situated within the Kruger National Park in South Africa. The evaluation is done for two soil depths namely the surface ( i.e., 0-10 cm) and root zone soil moisture (i.e. 10-100 cm) with the objective of understanding how the respective data products capture characteristic patterns of soil moisture within a 25 km x 25 km grid boxes that enclose each of the study sites. The evaluation includes an assessment of qualitative features of long term (i.e. multi-year), and short term (i.e., monthly) averages of the soil moisture signal relative to in situ measurements from each of the two flux tower sites. We learn that generally all the soil moisture products at all depths present higher magnitudes of soil moisture compared to observations, except for the CCAM-CABLE simulation output at the Malopeni flux site, which is closer to observations in magnitude. The difference in magnitude may be attributed to difference in spatial scale between in situ measurements and the rest of the products. The study therefore placed much focus on features of the soil moisture signal which may be attributed to as responses to the influence of the weather systems or climate variability of the region. The coefficient of determination ($R^2$), however reveals that most of the soil moisture products for the sites have an appreciable level of similarity (mostly $R^2 > 0.5$) at all depths. A qualitative analysis of the time averaged soil moisture signal, for all the products indicates that satellite observation, and satellite based model estimates capture most of the inter-annual structure of the soil moisture signal. We also learn from this study that all GLEAM models compare well with the in situ observations in reflecting the seasonality of soil moisture. It is therefore recommended that satellite derived model estimates can be used as surrogate observations for verifying the simulations of process-based models such as the CCAM-CABLE model. In particular, this can be done for instances where inter-annual variability as well as seasonal patterns of the soil moisture are of particular interest.

The CCAM-CABLE model simulation realistically represents the seasonality of the soil moisture cycle for the sites. However, the model fails to reflect some details such as the local soil-moisture maximum that occur in April at Malopeni. The simulation's strength in reflecting the changes in soil moisture across seasons demonstrates that it could be used to test the implications of long-term land cover changes or climate change on soil moisture patterns.

The study also investigated the level of uncertainty between GLEAM models and the CCAM-CABLE simulation. In particular a wavelet analysis was used to reveal, at a qualitative level, how periodic features of the soil moisture signal compare between the CCAM-CABLE simulation and the estimates produced by GLEAM models. In this case, the emphasis is on evaluating the extent to which both approaches have a joint variation or shared MI. The analysis has successfully revealed that both the simulation and model estimates equally reflect the periodic seasonal pattern of soil moisture, however there is a predominant time lag between GLEAM products and CCAM-CABLE. The time lag is of a time scale not exceeding a month at all soil depths (i.e., it lies between 5 and 20 days) during the studied years (2001-2014). We conclude that the major difference in the long and short term feature of the soil moisture signal, between CCAM-CABLE and GLEAM models estimates can be attributed to among other factors, their difference in capturing intra-annual patterns of the soil moisture signal. This is also supported by the existence of a non-negligible level of uncertainty on the onset and offset of the wet period which is calculated for the CCAM-CABLE and all GLEAM models outputs.

Despite the existence of uncertainly, we affirm that there is appreciable consistent information on the soil moisture signal from the simulation and GLEAM models. This is also reflected by the regional patterns of the SSI computed for each soil moisture product, and the MI between the CCAM-CABLE and GLEAM-v3c signal. The SSI values also reflect that soil moisture vary spatially from year to year. We demonstrated through the count of dry days that the index can be used to quantify soil moisture patterns at both short and long time scales across different landscapes. Future studies will explore patterns of the index on climate time scales. The spatio-temporal variability in the SSI can potentially be explained by climate systems that drive changes in soil moisture at different landscapes. Looking at the spread of the MI values within the study region, as well as their associated median values as grouped by soil types, it becomes evident that the extent of the shared features is not limited to the seasonal time frame.

The difference in the soil moisture signal structure at inter-annual time scales between the simulation and GLEAM models, opens-up an interesting question relating to the extent to which the influence of different drivers of soil moisture is represented by the simulation and estimation approaches. To understand this, future research will benefit from investigating the influence of changes in soil moisture drivers, particularly change in vegetation cover and soil type, on soil moisture memory. It will also be interesting to unearth the effects of extreme weather and climate change induced patterns on the long-term soil moisture pattern persistence. In this regard, it would be interesting to uncover the tipping or breaking points of trends in soil moisture. To this effect, we find the CCAM-CABLE and GLEAM representations of soil moisture patterns worthy of further investigation using various statistical approaches, including machine and deep learning algorithms, to gain a deeper understanding of soil moisture response to climatic and land management related effects.

*Team list and Author contribution*:

- Floyd – Developed research questions, analysed the data and compiled the manuscript.
- Marna and Mohau –suggested datasets to be explored, reviewed the manuscript, and made inputs on data analysis approaches and research questions formulations.
- Gregor – Inputs into the formulation of research questions, provision of in situ data, and critical discussion and review of the manuscript.
- Francois – Led the CCAM-CABLE model simulations and introduced the lead author to the model structure and the dynamical downscaling methods.
- Michael – Supervisory role and manuscript review.

*Competing interests*: The authors declare that they have no conflict of interest.

*Disclaimer*:

*Acknowledgements*: This work was funded by the EEGC030 project of the CSIR. The authors wish to acknowledge Ms Humbelani Thenga and Dr Marc Pienaar for their contributions.

*Data availability*:

**In situ data**

Data from the CSIR owned flux tower (i.e. Skukuza and Malopeni) can be requested from Ms Humbelani Thenga (HThenga@csir.co.za).

**ESA CCI**

The data are available online from www.esa-soilmoisture-cci.org/

**GLEAM**

The data are available online from www.gleam.eu

*Analysis scripts:*

https://github.com/vkhosa/Floyd-Vukosi-Khosa-HESS-2018.git

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

Appendix

## 5.1 Appendix A – Short term monthly inter compariason

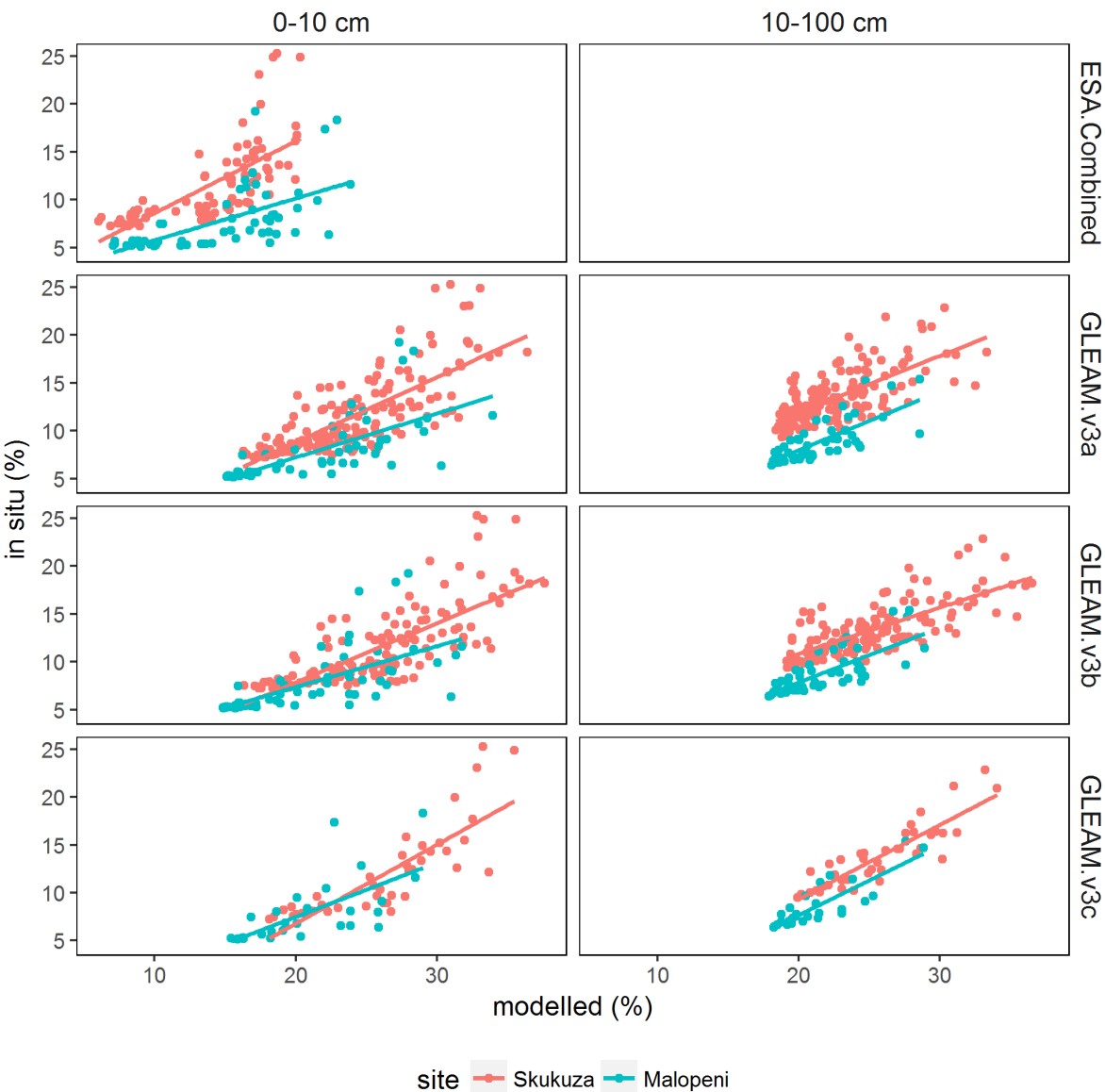

**Figure A1.** Comparison of monthly modelled and satellite products (red dots) with CCAM-CABLE (blue dots) surface (0-10 cm), and root zone (10-100 cm) soil moisture at the a) Skukuza (2001–2014) and Malopeni (2008–2013) sites respectively.

## 5.2 Appendix B – Cross wavelet analysis

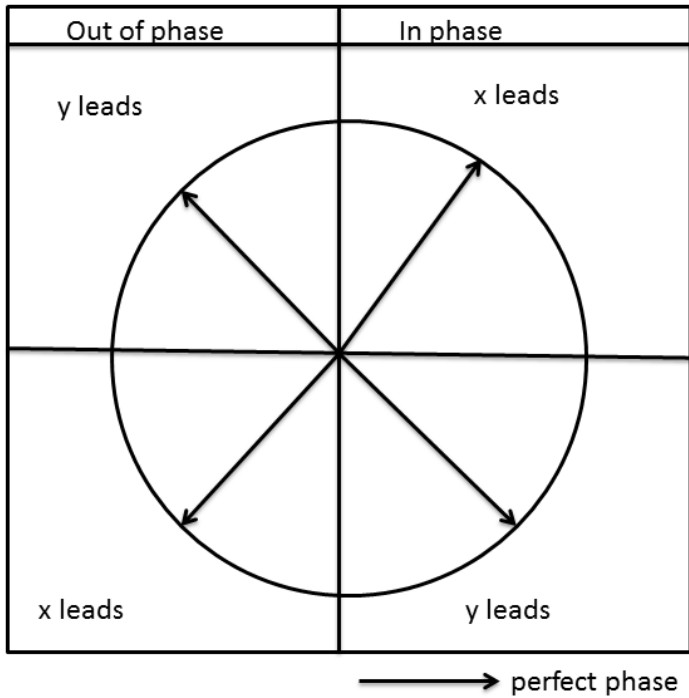

**Figure B1.** Phase interpretation between two time series $x$ and $y$. When series $x$ leads, $y$ lags and vice versa. This figure is inspired by a study by (Rosch and Schmidbauer, 2018).

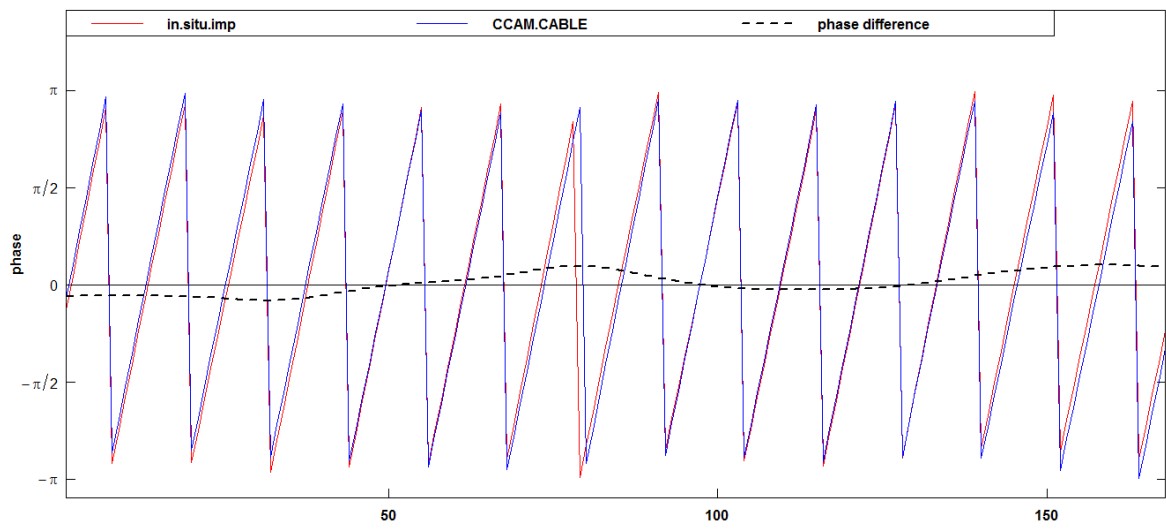

**Figure B2.** Phase difference between surface soil moisture simulated using CCAM-CABLE, and GLEAM v3a at Skukuza between 2001, and 2014 at period 12 at the surface.

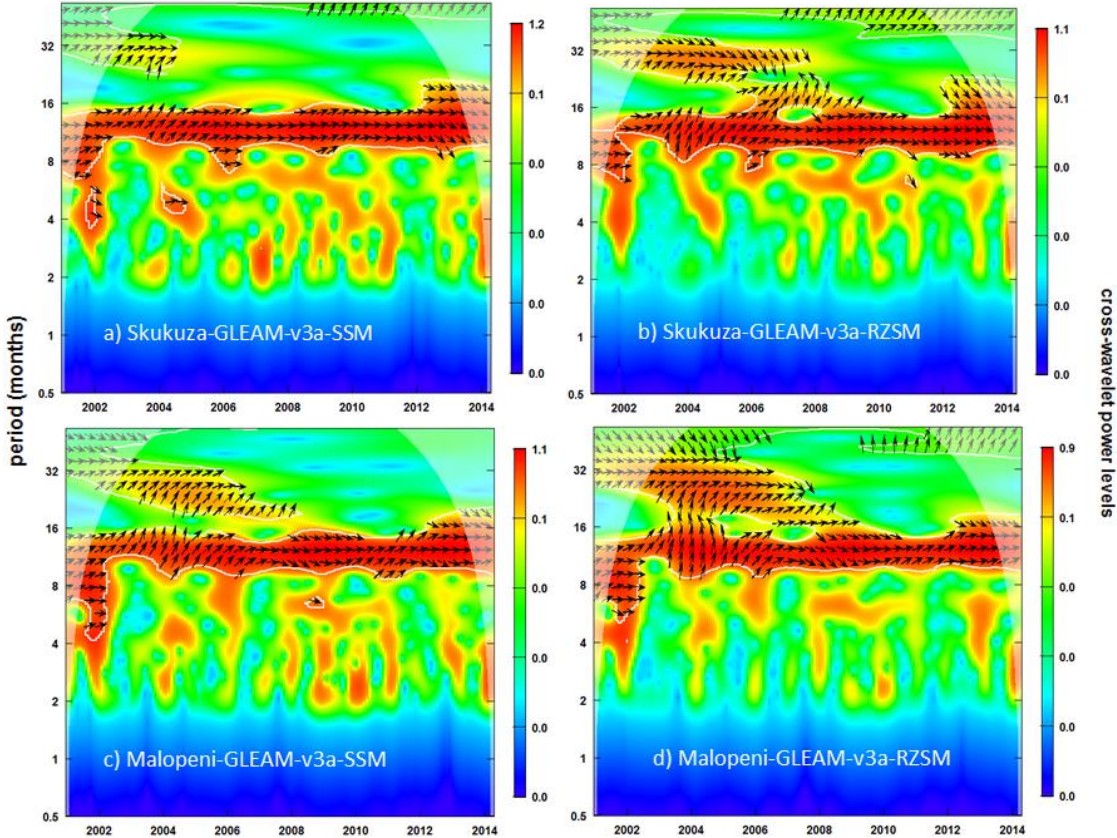

**Figure B3.** Cross wavelet power spectrum of surface (SSM, 0-10 cm) and root zone (RZSM, 10-100 cm) soil moisture between CCAM-CABLE, and GLEAM v3a at Skukuza (a, b) and Malopeni (c, d) respectively. The white contour lines indicate periods of significance at 10 %. The arrows pointing to the right indicates that the models are in phase, anti-phase point left, CCAM-CABLE leading GLEAM v3a is indicated by arrows pointing straight down. The dome shape (shaded areas) represents the cone of influence between 2001 and 2014.

## 5.3     Appendix C – Multiple imputation

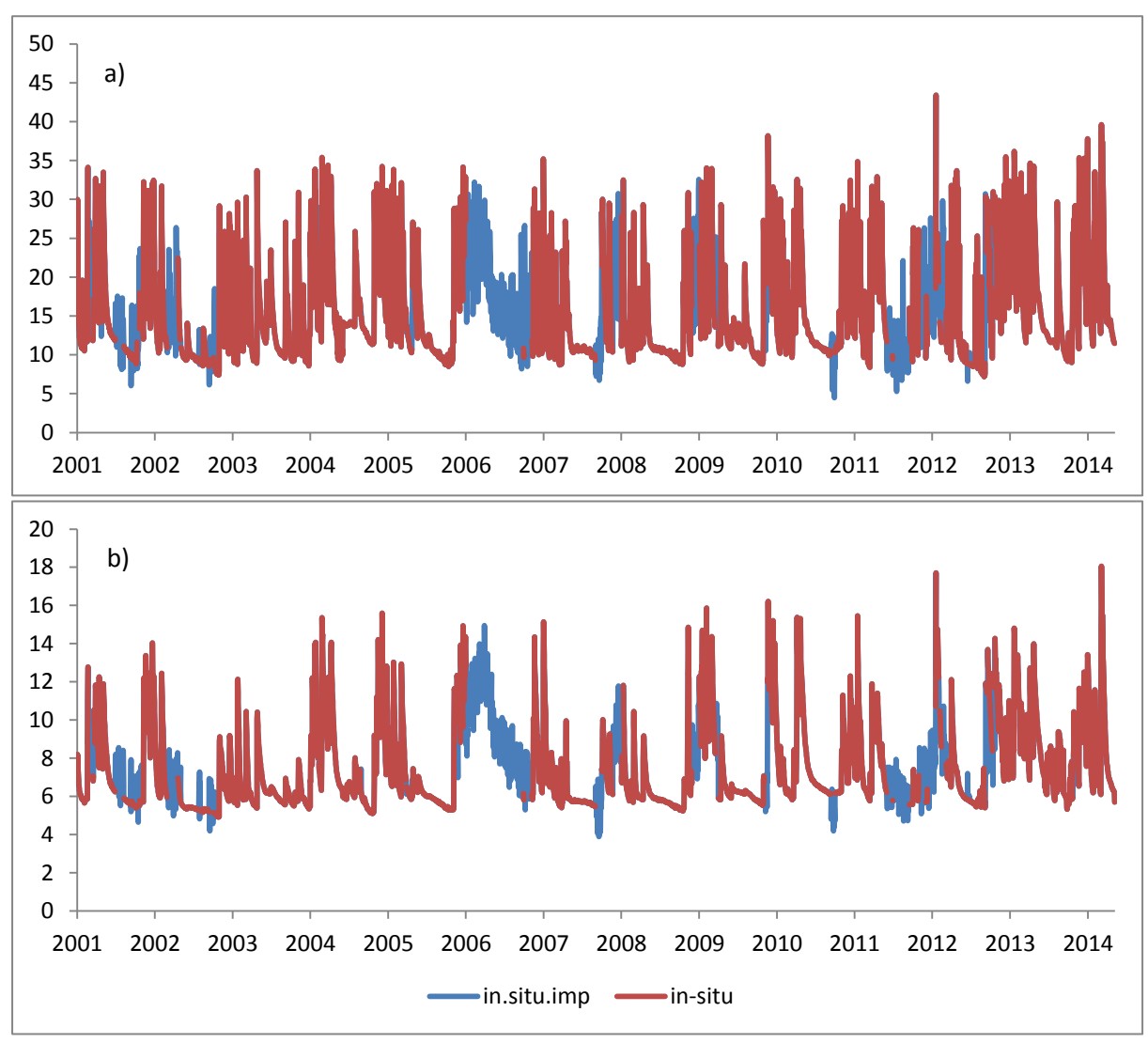

**Figure C1.** Daily a) surface and b) root zone soil moisture time series at Skukuza showing the imputed parts (blue) of the time series and the observed parts (red).

**Table C1.** Statistics of the distribution of the imputed and observed time series of surface and rootzone soil
moisture at the Skukuza site.

| Surface soil moisture | Original data | Imputed data |
|---|---|---|
| Mean | 15.59 | 15.76 |
| Median | 13.33 | 13.83 |
| Standard deviation | 6.21 | 6.10 |
| Variance | 38.68 | 37.22 |
| **Root zone soil moisture** | | |
| Mean | 7.45 | 7.55 |
| Median | 6.49 | 6.69 |
| Standard deviation | 2.18 | 2.17 |
| Variance | 4.76 | 4.74 |