# Peer review of "Evaluation of soil moisture from CCAM-CABLE simulation, satellite-based models estimates and satellite observations: Skukuza and Malopeni flux towers region case study"

_Hydrology and Earth System Sciences, 2018_

## Referee Comment (RC1) · Anonymous Referee #1 · 28 Nov 2018

This study is focused on evaluate soil moisture estimations from CCAM-CABLE and GLEAM outputs as well as ESA-CCI product using in situ observations. The writing is good and the paper is readable. While my primary concern is that soil depth for in situ observations is not well consist with either model- or satellite-based soil moisture estimations. Uncertainties from the preprocess as Equ(1) are hard to be assessed, due to propagating surface soil moisture information to deeper soil layers is a very complex procedure and relies on such as soil texture. Given these artificial errors, readers may hard to build their confidence in this study.

[Figure]

Minors:

1. The authors indicate that this is due to lack of publically available in situ observations in Africa (Lines 81-83, Lines 130-131), yet at least International Soil Moisture Network may provide more abundant observations in Africa. Thus, the authors may want to narrow their study in South Africa, and then revise the related introductions accordingly.

2. Data availability for ESA-CCI product is very low before 2008. While Coverage fractions for model-based simulations are basically 100%. Will data availability differences have impacts on the results?

3. Line 251, given daily (even finer temporal resolution) satellite and model soil moisture, why evaluations are focused on monthly time series?

4. Lines 191-195, indeed ESA-CCI is not the unique blended soil moisture product. The Soil Moisture Operational Products System (SMOPS), for example, also provides an operational global blend of all available microwave soil moisture retrievals on a daily basis (Yin et al., 2015).

5. Section 2.2: which version ESA-CCI data was used in this paper? Line 174-175, CDF-matching to what? Lines 179-186, passive observations are based on radiometer, while it does not indicate passive sensors are only able to take measurements during daylight hours. Besides, whether satellite signals may penetrate clouds fog, vegetation mainly rely on wavelength, rather than what kind of sensors (Wang et al., 1987; Jackson et al., 1989; Wagner et al., 2013).

Kerr Y. H., P. Waldteufel, J. -P. Wigneron, et al. The SMOS mission: New tool for monitoring key elements of the globalwater cycle. Proceedings of the IEEE, 2010, 98(5), 666–687. Wang J R, E T Engman, T Mo, T J Schmugge, J C Shiue. The effects of soil moisture, surface roughness, and vegetation on L-Band emissions and backscatter. IEEE Trans. Geosci. Remote Sens., 1987, GE-25(6): 825–833 Jackson TJ, Schmugge TJ. Passive microwave remote sensing system for soil moisture: some

supporting research. IEEE Trans. Geosci. Remote Sens., 1989, 27(2): 225–235 Yin J., X. Zhan, Y. Zheng, J. Liu, L. Fang, and C. R. Hain. Enhancing Model Skill by Assimilating SMOPS Blended Soil Moisture Product into Noah Land Surface Model. J. of Hydrometeorol., 2015,16(2): 917-931

---

## Referee Comment (RC2) · Anonymous Referee #2 · 30 Nov 2018

The manuscript tries to compare soil moisture estimates from a coupled GCM-dynamic land surface model, and from 3 variants of passive/active remote sensing product (25km resolution) against small scale in-situ measurements (2/4 profiles a 4 depths) from two South African flux tower sites. Measurements and Estimated were compared on aggregated monthly time steps, for 2 depths (0-10, 10-100cm) looking at the correlation between monthly time series, phase lags analysed by wavelet analysis and representing the on and offset of a wetting period. In general, I belief the topic and analysis presented is of interest to the readership of HESS and could therefore be

considered for publications. However, I have some general as well as some specific comments that would need to be considered and require major revisions before possible publication.

General Comments

1. As a hydrologist, I am interested in the daily (sometime even hourly) dynamics of the individual components of the water cycle. As a water manager, we have to provide runoff predictions on hourly to daily timescales to hydropower producers or to release warnings on flood and low flow conditions. The question for me is, to what extend is a soil moisture estimate relevant that compares on a monthly level with an R2 of 0.5? What is the performance on daily estimates?

2. It seems that all products use different sources of precipitation input. How does the precipitation input differs and compares to the measurements of the two flux tower sites. I assume that at least some of the deviations in the soil moisture dynamics stem from differences and deviation in the precipitation dynamics.

3. The same hold for temperature, humidity and other inputs used for ET-calculation.

4. How representative are the averaged soil moisture data for the 25*25km2 pixels. My experience is that soil moisture data largely vary in space with short correlation lengths. What is the variation in soil texture over the 25*25km2 domain? I still see a large gap in scale that at least has to be discussed.

5. How do temporal difference in soil moisture behave of different time scales (days, weeks, months)? Perhaps that is an information, which is more similar covered by all products/estimates.

Specific Comments

P2l49ff It should be mentioned that soil moisture itself is not the driving force for water transport and evapotranspiration, rather it is the soil matric potential. Often difference in soil moisture only reflect differences in soil texture.
P3l96: Could you explain data constrains more precisely. My experience is, that there are hundreds of FLUXNET locations available, some of them also providing soil moisture data. So I do not see that you are limited or constrained by available data!

P5l179ff: As you are using a product combining active and passive microwave data, I would be more specific here. Passive microwave by the way is not dependent on radiation, it emits dependent on it temperature and emissivity. Also, active sensors per se are not necesseraily able to penetrate through vegetation – this will be largely dependent on the wavelength (x-,C-,L-band). Please be more precise on that topics.

P8l293: Why is the focus more on the phase agreement rather than on the magnitude? Because the results are better!? Or because it is more important!?

P9l317: Why are only detrended data analysed? If there are trends that are different, this would be interesting as well!

P12L390ff: I feel that large parts of the discussion would benefit from some short introduction of how the different products are generated (e.g. GLEAM, built on Priestley & Taylor, Stress-function bsed on VOD derived from mircrowave products . . .). In its current form some of the discussions remain relatively week.

P20/21 Why is cov used in Fig. 10 and 11. As I do not know the Standard deviation the correlation coefficient would be more intuitive for me!?

P22l688: Should readers really be surprised by the conclusion that all products/estimates are at least able to reproduce the seasonality in the soil moisture signal! I am sure taking some mean monthly precipitation information, Temperature as a proxy for ETp and some simple bucket model would provide some similar performance. I know this is provocative, but my impression is you should at least demonstrate that all the effort you are doing is significantly better than such a Null-model!

Minor comments

P1l3: should be ". . . satellite based model estimates"

P1l20: should be ". . . turn out"

P7l263: semicolon should be removed

P8l283: should be "inter-compares"

P12l408ff: which figure is this text referring to?

P11l378: how are "wet periods" defined?

P18l581: structure of the sentence

P20l728: What you mean by soil moisture memory!

---

## Author Comment (AC1) · 25 Jan 2019

1. "As a hydrologist, I am interested in the daily (sometime even hourly) dynamics of the individual components of the water cycle. As a water manager, we have to provide runoff predictions on hourly to daily timescales to hydropower producers or to release warnings on flood and low flow conditions. The question for me is, to what extend is a soil moisture estimate relevant that compares on a monthly level with an $R^2$ of 0.5? What is the performance on daily estimates?"

[Figure]

We agree with the reviewer that for application purposes it is desirable to have predictions at finer temporal scales such as hourly or daily. In principle, process-based models such as CCAM-CABLE, which is also one of the discussed models in this paper, could be configured for such short time scale operational mode runs. The success of such a set-up depends on the forcing data in which case it becomes important to force the models with the observed climate states. Such short time scale investigations are outside the scope of this paper. The investigated setup for CCAM-CABLE simply dynamically downscale ERA interim forcing data from a 50 resolution to a to 8km resolution leading to soil moisture estimates at the same resolution. The R2 value reflects uncertainty between the models at the analysed time scales. A sizable effort is expanded in this regard to highlight aspects of model uncertainty (i.e., L548-558 and L624-626) some of which alludes to the assumptions about homogeneity of vegetation and hence soil texture classes in these models. A full quantification and attribution of model uncertainly is indeed a topical issue and deserves a separate treatment.

2. "It seems that all products use different sources of precipitation input. How does the precipitation input differs and compares to the measurements of the two flux tower sites. I assume that at least some of the deviations in the soil moisture dynamics stem from differences and deviation in the precipitation dynamics.", "The same hold for temperature, humidity and other inputs used for ET-calculation."

As stated on the manuscript in L104-124 the goal is to compare models estimates within situ observations, particularly in capturing the seasonal cycles of soil moisture for local conditions to uncover strengths and weaknesses of the various products. Models evaluated range from complex to simple with regard to structure.

3. "How representative are the averaged soil moisture data for the 25*25km2 pixels. My experience is that soil moisture data largely vary in space with short correlation lengths. What is the variation in soil texture over the 25*25km2 domain? I still see a large gap in scale that at least has to be discussed."

[Figure]

We agree with the reviewer that soil moisture is highly variable over space and time. This issue of scale is often found to be debatable in field. The models make a homogeneity assumption per grid-box for most meteorological and environmental drivers such as temperature, vegetation and soil texture types within a chosen grid scale. Clearly, within, this model assumption, soil moisture signal averaged over a monthly time scale, yields an effective pattern. Point comparison, or multi scale, between observations and model outcomes mostly are mostly assumed to be interpretable when the homogeneity assumption are consistent with the site specific details on drivers. In particular, a well-developed signal at small length scale, by deduction, may be deemed representative of a larger region belonging to the same climate system and having similar drivers.

4. "How do temporal difference in soil moisture behave of different time scales (days, weeks, months)? Perhaps that is an information, which is more similar covered by all products/estimates."

On very short times scales such as hourly to daily time scales, local effects can lead to a pronounced noise of the observations however such noise is anticipated to lead to compensating effects upon long term averaging. In this paper we focus on much longer time scales when the soil moisture signal is well developed.

Specific Comments

"P2l49ff It should be mentioned that soil moisture itself is not the driving force for water transport and evapotranspiration, rather it is the soil matric potential. Often difference in soil moisture only reflect differences in soil texture."

Thank the reviewer for highlighting that soil matric potential is an important driver for water transport. Attributing it as the sole driver as the reviewer suggests might obscure the fact that there are other driving factors such as gravity, potential energy, capillary forces and hydraulic activity as discussed in Bonan (2008). In response to the reviewers comment, we will update the manuscript with a comprehensive list of all dominating

drivers of water transport and evapotranspiration.

"P3l96: Could you explain data constrains more precisely. My experience is, that there are hundreds of FLUXNET locations available, some of them also providing soil moisture data. So I do not see that you are limited or constrained by available data!"

In terms of the study domain, our high-resolution domain covers northeastern South Africa, including Kruger Park. Understanding the soil-moisture over the semi-arid regions of Kruger Park where we have the flux towers is important from a conservation perspective. Moreover, northeastern South Africa has a strong ENSO signal (Engelbrecht et al., 2011) and understanding soil-moisture in this region is important in terms of understanding the impacts of climate variability and change on agriculture, live-stock production, biodiversity and thus also tourism. Unfortunately, for this region of interest there is only one station reporting to FLUXNET, which is the one used in this study. To this effect, we would like to humbly reiterate to the reviewer that FLUXNET station density is indeed seriously constrained for studies on soil moisture, including the present study.

"P5l179ff: As you are using a product combining active and passive microwave data, I would be more specific here. Passive microwave by the way is not dependent on radiation, it emits dependent on it temperature and emissivity. Also, active sensors per se are not necesseraily able to penetrate through vegetation – this will be largely dependent on the wavelength (x-,C-,L-band). Please be more precise on that topics."

We thank the reviewer on pointing the differences between passive and active sensor, we will precisely discuss these in the updated manuscript.

"P8l293: Why is the focus more on the phase agreement rather than on the magnitude? Because the results are better!? Or because it is more important!?"

Whereas it is appealing to evaluate the models on the basis of agreement in both magnitude and significance, it is instructive to focus on highly predictable part of a climate

system which can be bench-marked with an intuition. We focus much on phase agreement or seasonal cycles as these are intuitive features of the climate system which should be effectively predictable by models across the considered length scales. Further motivation of this point is highlighted on the manuscript P3L104-L117 and P7L261-266.

"P9l317: Why are only detrended data analysed? If there are trends that are different, this would be interesting as well!"

We agree with the reviewer that analysing trends on the data would be interesting and we think it deserves a special attention in its own right. However, we do not see how such an analysis can fit within the scope of the present study. Our calculation of the covariance is geared towards depicting the extent of mutual information among the respective models. The underlying statistical assumption for the calculation of covariance is that the input data should be stationary, thus the detrended and deseasonalised time series data is purely dictated by the standard procedure for obtaining the desired statistics.

"P12L390ff: I feel that large parts of the discussion would benefit from some short introduction of how the different products are generated (e.g. GLEAM, built on Priestley & Taylor, Stress-function bsed on VOD derived from mircrowave products : : :). In its current form some of the discussions remain relatively week."

We thank the reviewer for raising this point. For clarity, the manuscript will be revised by reiterating, in the in the discussion section, some of the information highlighted in the introduction section on how different products are generated.

"P20/21 Why is cov used in Fig. 10 and 11. As I do not know the Standard deviation the correlation coefficient would be more intuitive for me!?"

It makes perfect sense to choose correlation as a measure of similarity between variables over covariance, in the case when effects of the change in location and scale

are not of special interest. A covariance, is utilised in this case as an analogue of mutual information between respective models because its covariance is expressed in units that vary with the data in which case we can also see location and scale induced variation.

"P22l688: Should readers really be surprised by the conclusion that all products/estimates are at least able to reproduce the seasonality in the soil moisture signal! I am sure taking some mean monthly precipitation information, Temperature as a proxy for ETp and some simple bucket model would provide some similar performance. I know this is provocative, but my impression is you should at least demonstrate that all the effort you are doing is significantly better than such a Null-model!"

Presented in this paper are models of with differing construction. The aim, as explain in the introduction section, is to evaluate their performance against well understood seasonal patterns of the system as reflected by in-situ data. Complex process-based models like CCAM-CABLE, as is the case in this study, simulate a climate systems through coupled atmospheric and terrestrial processes modules which exchange information at run time. It is not given that these models should be able to predict climate system patterns at policy relevant length scales as these depends on several factors including the model sensitivity to the forcing data. In the case that such complex models demonstrate such predictive power, even for seemingly simple cases, their demonstrated utility and value stand a chance to translate to various computer experiments including those of climate change projections. In this regard the suggested bucket model is inferior in its assumptions and simplicity. A natural starting point on evaluating the value of process-based mechanistic models is on the predictable aspects of the climate system as demonstrated in this paper. It is not the interest of this paper to embark on inter-model comparison. We are afraid such a comparison, including that of bucket model as suggested, may not lead to conclusive insight. This point is also clearly articulated on the manuscript in L117-120.

Minor comments

"P1l3: should be ": : : satellite based model estimates"" We thank the reviewer for the comment. We will amend manuscript as suggested. "P1l20: should be ": : : turn out"" We thank the reviewer for the comment. We will amend manuscript as suggested .

"P7l263: semicolon should be removed"

We thank the reviewer for the comment. We will amend manuscript as suggested. "P8l283: should be "inter-compares""

We thank the reviewer for the comment. We will amend manuscript as suggested. "P11l378: how are "wet periods" defined?"

The wet period is defined as the summer or the rainfall period (i.e. November to April).

"P18l581: structure of the sentence"

We thank the reviewer for the comment. We will amend manuscript as suggested. "P20l728: What you mean by soil moisture memory!"" Soil moisture memory refers to the ability of soil to "remember" dry and or wet anomaly.

---

## Author Comment (AC2) · 25 Jan 2019

"While my primary concern is that soil depth for in situ observations is not well consist with either model- or satellite-based soil moisture estimations. Uncertainties from the preprocess as Equ(1) are hard to be assessed, due to propagating surface soil moisture information to deeper soil layers is a very complex procedure and relies on such as soil texture. Given these artificial errors, readers may hard to build their confidence in this study"

The propagation of surface soil moisture to deeper soil layers is very complex and we fully acknowledge that soil texture plays a major role in the propagation of moisture to deeper levels. Equation 1 calculates a vertically integrated weighted mean soil moisture value using the thickness of the different soil layers (horizons) expressed as a ratio of the total depth. The soil texture changes throughout the soil profile at both the flux tower sites and we selected a slightly more advanced method to calculate a representative soil moisture value than merely calculating an arithmetic mean. "The authors indicate that this is due to lack of publically available in situ observations in Africa (Lines 81-83, Lines 130-131), yet at least International Soil Moisture Network may provide more abundant observations in Africa. Thus, the authors may want to narrow their study in South Africa, and then revise the related introductions accordingly."

Generally in situ data for soil moisture is lacking for most of the regions in Africa, especially in South Africa, this is evident from both the FLUXNET (with two sites) and International soil moisture network (ISMN) with no sites in South Africa and three sites on the African continent. However, we agree with the reviewer that the current study should be delineated for South Africa, as shown in figure 1. A revised version of this manuscript will clearly articulate this in the introduction as well. "Data availability for ESA-CCI product is very low before 2008. While Coverage fractions for model-based simulations are basically 100%. Will data availability differences have impacts on the results?"

The data is averaged to monthly, reporting only cases where the daily data availability is at least 80% . Furthermore, comparison between modelled data and in situ observations is only done for periods where 80% of in situ data is available (i.e., the periods that have missing data or <80% are not considered in the analysis). We acknowledge that the evaluation metric value will change when more data points are available. However, we are not intercomparing products (Figures 5 and 6) but we are evaluating how each product compares against in situ observations (Figure 7). We would like to refer the reviewer to lines 447-456 where we acknowledge the issue of different sample sizes

used to compute the evaluation metric. The results only provide an indication of how each product compares against the in situ observations.

"why evaluations are focused on monthly time series?"

On very short times scales such as hourly to daily time scales, local effects can lead to a pronounced noise of the observations however such noise is anticipated to lead to compensating effects upon long term averaging. In this paper we focus on much longer time scales when the soil moisture signal is well developed.

"Lines 191-195, indeed ESA-CCI is not the unique blended soil moisture product. The Soil Moisture Operational Products System (SMOPS), for example, also provides an operational global blend of all available microwave soil moisture retrievals on a daily basis (Yin et al., 2015)."

We thank the reviewer for this comment and for pointing us to other data products. The ESA-CCI datasets however, were selected as they currently presents highly data coverage spatially as it merges data from different satellites, individual data products are likely to have low data coverage spatially over time. "Section 2.2: which version ESA-CCI data was used in this paper? Line 174-175, CDF-matching to what? Lines 179-186, passive observations are based on radiometer, while it does not indicate passive sensors are only able to take measurements during daylight hours. Besides, whether satellite signals may penetrate clouds fog, vegetation mainly rely on wavelength, rather than what kind of sensors (Wang et al., 1987; Jackson et al., 1989; Wagner et al., 2013)."

Version 03.2 of the ESA-CCI dataset was used in this study. We thank the reviewer for raising this comment and we will clearly indicate the version of the data used on the updated manuscript. CDF is used to merge the data from active and passive sensors using vegetation cover as described in L177-178. We thank the reviewer on pointing the differences between passive and active sensor, we will precisely discuss these in the updated manuscript.

---

## Author Response (AR1)

Response to Editor's comments

Dear Editor,

We thank you for the comments raised on the submitted manuscript. We have accommodated the suggested modifications as outlined below. Our responses to the comments follow the comments made (repeated in quotation marks), followed by our responses in bold. Below is a list point-by-point response to the comments raised on the previously submitted manuscript.

"What information can be expected from an averaged soil moisture (in lateral space, over depth and over time) in general?"

- **Whereas soil moisture has a short correlation length, averaging soil moisture over a profile can yield valuable information concerning month-to-month variability in soil moisture and the associated seasonal cycle. A study which largely inspired the analysis for this paper, presented related information from a number of sensors at various soil depths. See for example the study by Yuan and Quiring (2017)**

"What added information or reliable estimate can be derived from the respective sources? Or in other words what is the trivial part which can be gained by much simpler means like the water balance or distributed linear stores etc.?"

- **Assuming "respective sources" in the above comment refers to the data sources (e.g., modelled, satellite, satellite-based modelled data sources used) then we anticipate that: 1) Testing the ability of the different sources of data to capture the soil moisture dynamics as driven by known (in some cases quantified) as well as unknown (and therefore also unquantified) drivers of the fluctuations in soil moisture at the study area and study sites. 2) We acknowledge that the water balance/distributed linear stores approach can provide a simplified way of capturing average seasonal effects and rainfall-soil moisture relationships. The more complex land surface model (coupled land surface-atmospheric model), that is presented here, capture some of the key processes which drive fluctuations in soil moisture such as vegetation. This presents an opportunity to investigate if the modelled drivers can yield patterns that are representative of the SM over time. For practical purposes: if a simple model (e.g., water balance model) is more accurate than the more complex models and satellite observations then one will use the simpler model. One of the goals of the paper is to evaluate if these data sources, and more specifically the process based model CCAM-CABLE output is reliable or accurate for the specific area (latitude/longitude) having a specific climate condition (semi-arid) and ecosystem (savanna). We attempt to establish if the models in particular CCAM-CABLE can be used at all to simulate local and regional soil moisture conditions for the selected resolution (spatial and temporal). We conclude on page 29 (paragraph 785-790) with regard to using satellite derived SM or process based models (i.e., CCAM-CABLE). Regarding that all these models are limited in their assumption about the drivers of soil moisture, the evaluation output can potentially inform their fitness for purpose and potential avenues for improvement.**

"Given your expertise and previous publications on the method, I suspect that allowing a slightly more clear focus of the manuscript towards the actual topic of the special issue would be beneficiary to both - the manuscript itself and the contribution to the special issue."

- **We thank the Editor for the recommended additions, in order to improve the manuscript with this regard and align it with the special's issue actual topic, the standardised soil moisture index (SSI) is computed and used to analyse spatial and temporal as well as inter-model variations of the simulated soil moisture. The calculation demonstrate a way of establishing a link between the soil moisture states and topography, and hence landscapes. This additional discussion on landscapes is described in Sect. 2.4.4 and the results are presented in Sect. 3.2.**

"You summarise that the coherence of CCAM-CABLE and GLEAM results indicate "that the key physical processes that drive soil moisture in CCAM-CABLE and GLEAM, at the surface and root zone, lead to an appreciable degree of mutual information." Since mutual information is well-defined in information theory, I would expect some more detailed reference and calculus what information is shared and conveyed in the different approaches. More importantly, having a strongly seasonal input signal, I would expect quite substantial shared information between monthly averages of soil moisture and rainfall. So what information is added by the models then?"

- **We thank the editor for the suggestion, to address the above comments, in the updated manuscript we compute mutual information using Shannon entropy as described by Kraskov et al. (2004). Detailed calculus and references in relation to the calculation are outlined in Section 2.4.5 of the manuscript.**
- **A calculation of mutual information as part of model evaluation, gives a perspective on the extent to which different soil moisture models captures similar processes on a grid cell as driven by variables such as precipitation and temperature.**
- **A comparison of rainfall and soil moisture signal at Skukuza and Malopeni on Figure 2, reflects that there could be a time lag between the observed soil moisture and accumulated monthly rainfall. Regarding that the models do differ in their assumption about the drivers of soil moisture, the mutual information calculation in this case becomes instrumental in diagnosing the extent to which differences in the respective model outputs are comparable.**

"As far as I understood, your reference data at Skukuza and Malopeni are both in sandy loamy soils. Your gridded soil map however, classifies the area near the stations as silty loam. If the soil types are important (as you claim), how does this affect your results? How does this relate to the covariance structures as estimate of uncertainty, when the covariance for the reference station sites is very low and other references are lacking? Are this topographic effects or patterns in the rainfall estimates or other compartments of the water balance? "

- **We thank the Editor for raising this comment. The gridded soil type data present a 25*25 km$^2$ grid cell indicating the dominant soil type. The dominant soil type at the grid cell is deemed representative of the entire grid cell thus introducing a soil type homogeneity assumption at a grid cell. The grid cell within which the flux towers are located is silty loam. However, at the flux tower foot print (i.e., 1 km$^2$) the soil type is found to be predominantly sandy loam. This is now clarified in the manuscript in Sect. 2.1.1 and Sect. 2.1.2 for Skukuza and Malopeni respectively. This however does not affect the interpretation of the mutual information presented at the spatial resolution of 25 km. Furthermore, instead of using the covariance mutual information is now computed using the Shannon entropy as discussed in Section 2.4.4 of the manuscript.**

- **A discussion on spatial patterns of the standardised soil moisture index is added and used to link to link the soil moisture spatial patterns to topographic features at the study region and hence the associated landscapes.**

"Having a strongly seasonal signal and rather limited reference data, I am not quite sure that I understood the value of employing the cross-wavelet analysis. I do not find it very surprising to see a significant annual period and I would not expect any shift in this frequency. What can one learn about the validity of CCAM-CABLE or GLEAM-v3a if there is no ground truthing data involved? What is the frequency spectrum of the reference time series? Maybe even the mean annual soil moisture

dynamics could serve as a reference when the inter-annual deviations are specified? Again, my question would link back to the information contained in the different sources under study."

- **We thank the editor for the recommendations. To address this comment, we resorted to using observed data for the cross-wavelet analysis. To achieve this, we gap-filled the in situ observations using the multiple imputations method. This is unavoidable regarding that the cross-wavelet analysis only works with complete datasets. Gap filling was only employed for the Skukuza since it has the least gaps as compared to Malopeni. As a result, the cross-wavelet analysis is now computed between CCAM-CABLE, GLEAM-v3a and in situ observations both at the surface and rootzone, thereby presenting an opportunity to evaluate the time lag and cyclic features of the modelled soil moisture signal. Furthermore. The gapfilled dataset is also used in the simulation of the onset and offset of the wet period analysis. The two analyses mentioned are carried out for the years ranging between 2001 and 2014.**

"Moreover I find it rather difficult to trace the respective reference data throughout the manuscript. Since you argue about using flux station data, I suspect that the water and energy balances can be closed with the available data. However, you only compare the most difficult to evaluate part of the water balance (soil moisture). Would there reside as much information about the respective states and processes in ET, P and Q? How do these compare to the downscaled GCM and GLEAM inputs?"

- **A detailed analysis on ET has been done on a separate manuscript currently under review. Conducting a water balance study would be interesting. However, certain components of the water balance such as drainage and runoff are not measured at the sites.**
- **To maximise the use of the reference datasets, we gapfilled the in situ observations as mentioned above. This gap filled datasets enabled cross-wavelet analysis and the computation of the onset and offset of the wet period which is only possible on a complete dataset. However, the Malopeni datasets would still not feature in these analyses due to large gaps in the data, imputation of such datasets is likely to result in unreliable estimates of soil moisture.**

"I find it rather difficult to extract information from Figure 5 and 6. Since you are using an R2 for evaluation, maybe a respective Observed vs. Modelled dotty plot would be more insightful about the fit? How about further measures of correlation? Figure 6 is particularly difficult to evaluate. The lack of data appears to lead to overly positive rankings of correlation (in Fig. 7)."

- **We thank the Editor for the recommended improvements. Figures 5 and 6 are now represented quantitatively using the $R^2$ values plots (Fig. 3). However, the dotty plots of Observed vs. Modelled as recommended by the Editor are also presented in Appendix A.**
- **We agree with the Editor that the lack of data at the Malopeni site may lead to high correlations. This is expected if the number of data points that are compared are following a similar pattern.**

"Coming back to the overall manuscript and the proposed major revisions, I also suggest to include the figures along the lines of argumentation. Fig. 1 might benefit from some topographic information instead of politic boundaries. Fig. 2 did not give me much insight, which might be due to my scepticism to the suitability of the employed method. Fig. 3 and 4 might be combined?"

- We thank the Editor for the recommended improvements:

- **Fig. 1 – has been updated as suggested by the Editor, to include topographic information (i.e. altitude).**
- **Fig. 2 – has been moved to the appendix for description purposes.**
- **Fig. 3 and 4 have been combined as recommended by the Editor.**

"With regard to your author comments to the reviews, I found it relatively difficult to grasp your intended lines of revision. I sincerely hope you will find my comments helpful in this respect. Please also note that both reviewers indicated room for improvement especially for scientific quality and scientific significance of the manuscript."

- **A number of revisions has been made on the updated manuscript considering comments from both the reviewers and Editor. These include minor and major changes such as:**
    - **Clarifying the in situ data constraints as raised by the two reviewers (L102-L103), improved descriptions of the passive and active satellite sensors (Section 2.2).**
    - **The description of the CCAM-CABLE model has been improved to address some of the concerns raised by the reviewers (Section 2.3.1); the improvements are contained in L204-216.**
    - **Motivation for using monthly data in the analysis is included in L257-260.**
    - **We cite the reference for Eq.1 in L283-284.**
    - **To maximise the use of reference data, we extended the use of in situ data to the cross-wavelet analysis and the simulation of the onset and offset of the wet period. This has been archived by employing the multiple imputations procedure since these analyses require complete datasets. The description of the multiple imputation procedure is outlined in L243-352.**
    - **The covariance analysis is now replaced with mutual information described in Section 2.4.5 as recommended.**
    - **The standardised soil moisture index is now computed among the models and used to link soil moisture simulations to landscapes.**

"Since our special issue is an inter-journal one which opens the opportunity to publish related data in ESSD, I would like to encourage you to consider publication of the data and scripts either as appendix to your MS or as own data publication. As far I understood, the data you used is partly hosted on publicly available repositories already. However you like to see the HESS data policy for details: https://www.hydrology-and-earth-system-sciences.net/about/data_policy.html"

- **The links to the analysis scripts are now shared**.

Regards

Floyd Khosa